

# Comparing many-body approaches against the helium atom exact solution

**Jing Li[1,2], N. D. Drummond[3], Peter Schuck[1,4,5] and Valerio Olevano[1,2,6⋆]**

**1** Université Grenoble Alpes, 38000 Grenoble, France
**2** CNRS, Institut Néel, 38042 Grenoble, France
**3** Department of Physics, Lancaster University, Lancaster LA1 4YB, United Kingdom
**4** CNRS, LPMMC, 38042 Grenoble, France
**5** CNRS, Institut de Physique Nucléaire, IN2P3, Université Paris-Sud, 91406 Orsay, France
**6** European Theoretical Spectroscopy Facility (ETSF)

⋆ valerio.olevano@grenoble.cnrs.fr

## Abstract

Over time, many different theories and approaches have been developed to tackle the many-body problem in quantum chemistry, condensed-matter physics, and nuclear physics. Here we use the helium atom, a real system rather than a model, and we use the exact solution of its Schrödinger equation as a benchmark for comparison between methods. We present new results beyond the random-phase approximation (RPA) from a renormalized RPA (r-RPA) in the framework of the self-consistent RPA (SCRPA) originally developed in nuclear physics, and compare them with various other approaches like configuration interaction (CI), quantum Monte Carlo (QMC), time-dependent density-functional theory (TDDFT), and the Bethe-Salpeter equation on top of the $GW$ approximation. Most of the calculations are consistently done on the same footing, e.g. using the same basis set, in an effort for a most faithful comparison between methods.

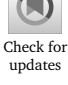
# 1   Introduction

The neutral helium atom and other two-electron ionized atoms are among the simplest many-body systems in nature. Here "many-body" is reduced to only three bodies, two electrons plus the nucleus. Even when treating the nucleus classically (i.e., as an external classical source and neglecting its wave function), in quantum mechanics two interacting bodies (the two electrons) already raise a *many-body* problem (an inhomogeneous two-body problem in helium). The Schrödinger equation cannot be solved in a closed form. The calculation of many-body correlation energies and correlation effects presents similar difficulties in two-electron systems (including the noteworthy case of the hydrogen molecule) as in any other many-body system.

Nevertheless, thanks to the pioneering work of Hylleraas in 1929 [1], the helium atom (and two-electron atoms in general) is almost a unique case in which we own an *exact* solution, though not in a closed form. By exploiting the full rotational symmetry of the system and rewriting the Schrödinger equation in reduced degrees of freedom, these being the three scalar Hylleraas coordinates over which the wave function is expanded as a power series, a numerical solution can be found. This numerical solution is "exact" in the sense that it consists of a number and a quantifiable margin of error on that number, together with the possibility of arbitrarily reducing that margin of error. The historical series of published results [2–11] (see Table I in [11]) has confirmed the numerically exact nature of the Hylleraas method for helium.

Hylleraas's original solution had a relative error of only $10^{-4}$, which is remarkable for a time in which computers did not yet exist. It played a fundamental role in assessing the validity of quantum mechanics as a universal theory that does not just apply to the hydrogen atom. Once higher-order effects are taken into account, such as nuclear finite-mass recoil (reduced mass of the electron and mass polarization term), relativistic fine structure (e.g. relativistic correction to the velocity, spin-orbit coupling, etc.), and quantum electrodynamic (QED) radiative corrections (the analog of the Lamb shift of hydrogen) [12–14], its quantitative agreement with the experiment, within the measured and calculated error bars, was one of the first triumphs of quantum mechanics [3].

Over the years, the Hylleraas calculation has been improved more and more [2–11], reaching an accuracy of 35 decimal digits in 2006 [10], a result confirmed and further extended [11], which required computer octuple precision. Beyond the academic interest, the comparison of such an accurate theoretical result with experimental measurements of the helium excitation spectrum has been proposed to estimate the fine structure constant accurately.

The availability of an exact solution suggests that the helium atom can serve as a workbench for many-body theories. Many body theories were at the beginning mostly tailored for systems with many, up to infinite particles. More recently one requires that a good many

body approach embraces simultaneously the small and high number of particles cases. A two-electron atom might appear a limiting case to study the many-body problem. However, it is not that far from the hydrogen molecule, of interest in molecular physics and quantum chemistry, or the deuterium nucleus, of interest in nuclear physics. Each of these systems presents a nontrivial many-body problem to describe the electronic correlation beyond the Hartree-Fock (HF) exchange. Many different formalisms beyond HF have been developed over time aimed at the solution of the many-body problem. While exact in principle, in practice all approaches rely on approximations and recipes whose validity are difficult to judge. The general tendency is to evaluate them against experiment. However, benchmarks against experiment are always affected by unaccounted effects not present in the theoretical description (non-Born-Oppenheimer, electron-phonon, relativistic corrections, etc.), which can mask the real many-body performances of the approaches. Validation of many-body approaches against the benchmark of an exact solution is an unavoidable step for further improvement. When calling for an exact solution one first thinks of a model system. Workbenches for many-body theories have been identified in more or less realistic simplified models, e.g. by replacing the long-range $1/r$ real electromagnetic interaction by a local interaction $\delta(r)$, and/or by discretizing the space, or somehow reducing the number of degrees of freedom of the system. One example of particular relevance here is the *spherium* model. With respect to the helium atom, in spherium only the angular degrees of freedom are considered, whereas the radial ones are dropped by confining the two electrons on the 2D surface of a sphere of radius $R$. However, the interaction is the real 3D $1/r$ across the sphere. $R$ is a model parameter which allows the tuning of the electronic density (like $r_s$ of the jellium model) and so of the correlations: this allows interesting studies which are impossible in real helium. By comparing the spherium solution to the real helium atom electronic structure (Fig. 2) one can appreciate the validity of this model to describe nature, as well as its limits. An interesting work on spherium also comparing many-body approaches is Ref. [15]. However the exact solution is often unknown even for simple models, or known only in particular cases or in reduced dimensions. Another drawback is that many-body theories could be checked on unrealistic features of models, and so one theory can be validated with respect to another on aspects that might be absent in real systems. So, we think that the helium atom and its exact solution is certainly preferable to more schematic models as a benchmark for many-body theories. Furthermore, the electronic structure of helium is very rich (see Fig. 2), presenting a complex spectrum of many excitations of different nature; it is certainly much more critical for a theory to be able to reproduce, as a whole, such a rich electronic structure rather than a model that can present just a couple of levels. Finally, the helium atom represents a very severe workbench for testing condensed-matter approaches devised for describing correlations in infinite solids by e.g. the introduction of the concept of "screening," a check that, according to our results, these approaches have surprisingly passed.

In the present work, the intention is to perform a comparison of several many body approaches. Most of those approaches are well known in condensed-matter physics. However, a direct comparison of their performances is often hampered by not consistent techniques of numerical resolution. One objective of the present work, therefore, is to improve on this. Second, we also want to introduce and apply a method used in nuclear physics which is the equation of motion (EOM) approach to go beyond the standard random-phase approximation (RPA). It is called the self-consistent RPA (SCRPA), of which the renormalized RPA (r-RPA) is a sub-product [16, 17]. We will give a short outline of this approach. We right now clarify that all along this paper we consistently use the nuclear physics convention to define the *random-phase approximation* (RPA) which in quantum chemistry and condensed-matter physics is rather known as linearized time-dependent Hartree-Fock (TDHF). The RPA here contains both the direct and the exchange terms, and should not be confused with the RPA of

condensed-matter physics (also known as the *ring* approximation), which only contains the direct term. We will here refer to the latter as dRPA (direct RPA) to avoid confusion. In order to provide the reader with an orientation table among the acronyms used in this article, in Fig. 1 we present the Feynman diagrams for the irreducible polarizability $\widetilde{\Pi}$ corresponding to all the approximations explored in this work. The last line of Fig. 1 presents the Dyson equation relating the irreducible $\widetilde{\Pi}$ to the reducible polarizability $\Pi$ whose poles are the excitation energies tabulated in this work for helium.

In order to compare with the other approaches, we at the same time calculate helium ground and excited states by some of the most widespread many-body approaches, including Hartree-Fock (HF), quantum Monte Carlo (QMC), quantum chemistry configuration interaction (CI), density-functional theory (DFT) and time-dependent density-functional theory (TDDFT), Bethe-Salpeter equation (BSE) [18–20] on top of the *GW* approximation [21–25], and the dRPA approximation on top again of the *GW* electronic structure, or also of the HF or the DFT ones (see Fig. 1). Some of these results were previously presented in the literature, but here we made the effort to recalculate most of them on the same footing, in particular using the same Gaussian basis set, which, as we will see, significantly affects the accuracy of the results. This yields a more faithful comparison between methods. [1] For obvious reasons, only the QMC calculations and the exact-DFT (including also TDDFT on top of exact-DFT) calculations, apart of course from the exact Hylleraas calculation, are not based on the Gaussian basis set. Most importantly, the spirit which has driven our comparison of so many methods was to understand and demonstrate the *effective* performances achieved *in practice* by a given methodology, avoiding idealistic statements. One can claim, for example, that "solving full Hedin equations self-consistently will provide the exact solution," but this remains an abstract statement if nobody was ever able to perform such a calculation for any real or even model system. When going for a real calculation one can face unforeseen problems related to, for example, basis-set issues, nonlinearity of equations, divergences to be avoided, self-consistency instabilities, etc., which can reduce the exactness of the solution achieved, if not actually preventing the achievement of a solution. We will discuss such issues in the present work. Beyond this, to situate the performance and pros and cons of each approach with respect to the others, the purpose of this article is to propose a workbench and a methodology for evaluation of future developments.

Most of the results shown in this paper are new and to our knowledge have not been presented earlier in the literature: i) r-RPA calculations have so far only been applied to models in use in nuclear physics and to the jellium-sphere model [16, 17], but not previously to real systems. We also give a short outline of this method, as it is not well known outside the nuclear physics community. ii) The novelty aspect is also apparent for our d-RPA calculations, which are applied on top of three approximations, namely HF, *GW*, and DFT-LDA. iii) Likewise for the *GW*, *GW*+RPA, and *GW*+BSE results studied as a function of starting point, from PBE [26] to full HF exchange (PBEh) [27]. iv) Although the DFT-LDA + TDLDA helium-atom excitation spectrum has been discussed several times in the literature [28–31], numerical results have never been published to our knowledge. We fill this gap in this work. v) Finally, our variational (VMC) and diffusion Monte Carlo (DMC) calculations present improved results with respect to earlier QMC works on helium, and the achievement of an accuracy high enough to be at the level of the experimental error bar.

---

[1]One should expect that, on localized gaussians, long-range $1/r$ methods (e.g., wavefunction based, GW, hybrids) converge more slowly than short-range $e^{-r}$ methods (e.g., LDA xc-potential or other DFT pure functionals) [85]: this is the reason why we chose basis-set families optimized for correlation wavefunction methods. Another possible way is to assess each method with its best complete basis set. However, if one checks Fig. 1 of Ref. [86], one can see that for He the convergence error can be reduced from $10^{-4}$ to $10^{-6}$ Ha using basis-set families optimized for DFT LDA. Since the error of the LDA approximation with respect to the exact Hylleraas is already $7 \cdot 10^{-2}$ Ha (Table 2), the basis-set convergence error of $10^{-4}$ can be neglected.

Figure 1: Irreducible polarizability $\widetilde{\Pi}$ in the various approximations studied in this work. HF+dRPA: direct RPA (dRPA) or ring approximation on top of the Hartree-Fock (HF) electronic structure; $GW$+dRPA: dRPA on top of the $GW$ electronic structure; DFT+dRPA: dRPA on top of the (either exact or approximated, e.g., LDA, GGA, etc.) DFT Kohn-Sham electronic structure; RPA (TDHF): random-phase approximation approximation, also known as linearized time-dependent Hartree-Fock; r-RPA: renormalized RPA; $GW$+RPA: RPA on top of $GW$; $GW$+BSE: Bethe-Salpeter equation on top of $GW$; TDDFT: linear response time-dependent density-functional theory with kernel $f_{xc}$ on top of either exact or approximated (e.g. LDA) DFT. Last line: Dyson equation $\Pi = \widetilde{\Pi} + \widetilde{\Pi} w \Pi$ between the irreducible $\widetilde{\Pi}$ and the reducible polarizability $\Pi$. The wiggly line marked $w$ indicates the bare many-body interaction (here the Coulomb interaction), while the double wiggly line indicates the screened interaction $W$.

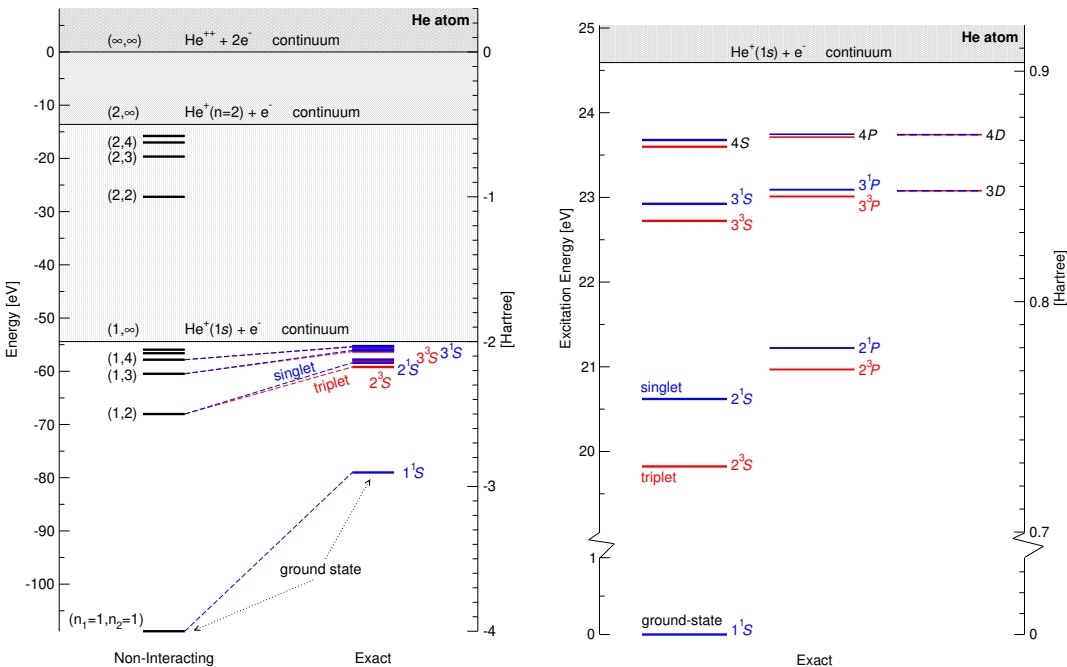

Figure 2: Helium atom full electronic structure (left panel). Both the noninteracting, independent-particle spectrum [Eq. (2)] and the exact [32] spectrum are shown. The right panel is a zoom on the first excitations of the exact spectrum.

The paper is organized as follows: we first introduce the electronic structure of helium and the exact Hylleraas solution, showing this to be a safe reference. We then describe in particular the SCRPA approach, referring to the literature for the other well known methods, and present the parameters of all our calculations. The results will be presented, first for the ground state and then for the excited states. Our conclusions are drawn at the end. We will generally use atomic units (Hartree, Bohr), but will also report energies in electronvolts (eV) when this is more intuitive. The zero of the energies will be fixed at the helium atom double excitation level $He^{++} + 2e^{-}$ when studying the helium ground state, and at the ground state $1^1S$ when studying the excitation spectrum.

## 2 Helium atom electronic structure and exact solution

The experimental spectrum of a real helium atom is affected by many effects (e.g. the finite mass of the nucleus, relativistic corrections, and QED radiative corrections) beyond many-body correlations. These effects are small corrections [3] that can be calculated at the first order, but must be taken into account in a comparison with experiment within experimental and theoretical error bars. Here we are interested in reproducing not the experiment, but an exact solution as a benchmark for many-body theories and their performances on correlation. So our workbench system will be an idealized nonrelativistic helium atom, with infinite nuclear mass and without relativistic and QED effects, whose Hamiltonian is

$$H = -\frac{\partial_{r_1}^2}{2} - \frac{\partial_{r_2}^2}{2} - \frac{Z}{r_1} - \frac{Z}{r_2} + \frac{1}{|\mathbf{r}_1 - \mathbf{r}_2|}, \qquad (1)$$

consisting of the kinetic terms, the interaction with the nucleus of charge $Z$, and the two-body Coulomb interaction between the electrons (last term). If we neglect the latter (noninteracting

or independent-particle approximation) the Hamiltonian can be split into two single-particle Hamiltonians of hydrogenic form, and the solution for the excitation spectrum can easily be written

$$E^0_{n_1 n_2} = -\frac{Z^2}{2}\left(\frac{1}{n_1^2} + \frac{1}{n_2^2}\right). \tag{2}$$

This is the noninteracting, independent-particle electronic structure reported (for helium $Z = 2$) in Fig. 2, left. One can identify the ground state, corresponding to the principal quantum numbers ($n_1 = 1, n_2 = 1$), the first excitations, $(1,2), (1,3), \ldots$, forming a Rydberg series up to the first ionization onset $(1,\infty)$ in which we are left with a $He^+(1s) + e^-$ helium positive ion in its hydrogenic $1s$ ground state, plus a free electron. We then have so-called double excitations ($n_1 > 1, n_2 > 1$), which are resonant with the continuum of the first ionization onset, and further single ionization onsets ($n_1 > 1, \infty$). Finally we have the full ionization level ($\infty, \infty$), in which we are left with the bare He nucleus plus two free electrons, $He^{++} + 2e^-$.

When comparing the independent-particle with the exact electronic structure (Fig. 2 left panel), one can see that the many-body term has an important, non-negligible effect already in helium. There are important shifts, especially for the ground state, and splits of levels according further quantum numbers as the total spin $S$ and the total orbital angular momentum $L$ (see also Fig. 2 right panel). A good many-body theory should be able to reproduce reasonably well both shifts and splits.

We now also briefly explain how the exact solution to the Schrödinger equation for the Hamiltonian in Eq. (1) could be obtained by Hylleraas. Starting from the solution of the hydrogen atom, and exploiting the full rotational symmetry of the ionic potential (an important simplification with respect to e.g. the hydrogen molecule) it was possible to write the electronic wave functions $\Psi(s, t, u)$ in terms of only three scalar coordinates,

$$
\begin{aligned}
s &= r_1 + r_2, \\
t &= r_1 - r_2, \\
u &= r_{12} = |\boldsymbol{r}_1 - \boldsymbol{r}_2|,
\end{aligned}
$$

instead of the two vectors or six scalars $\Psi(\boldsymbol{r}_1, \boldsymbol{r}_2)$ normally required for a two-electron system. The wave function is then written as a power series over the $s$, $t$, and $u$ Hylleraas coordinates,

$$\Psi(s, t, u) = e^{-ks} \sum_{l,m,n} c_{l,m,n} s^l t^m u^n, \tag{3}$$

apart from an important cusp factor $e^{-ks}$ in analogy with the solution of the hydrogen atom. It has been demonstrated [2] that the expansion Eq. (3), including negative powers $l, m < 0$, represents a formal solution to the He Schrödinger Eq. (1). The solution is found variationally, by minimizing the energy with respect to the free parameters $c_{l,m,n}$ and $k$. It is possible to select the symmetry of the wave function, for example by choosing even $m$ for space-symmetric singlet solutions and odd $m$ for space-antisymmetric triplets, like also the orbital character (upon reintroducing angular variables within multiplicative spherical harmonics [32]), and even the principal quantum number. This provides access not only to the ground state, but also all excited states, both energies and wave functions, and so also oscillator strengths. The Hylleraas accuracy of $10^{-4}$ (relative error), which was obtained with a reduced sum in Eq. (3) running only on positive powers, was in the following years improved by extending the series also to include negative powers [2]. An important increase in the accuracy was obtained thanks to a better description of the coalescence region at the origin by introducing a logarithmic singularity $\ln(s)$ [3], like in the wave functions which allowed Schwartz [10] to obtain an accuracy of 35 decimal digits,

$$\Psi(s, t, u) = e^{-ks} \sum_{j,l,m,n} c_{j,l,m,n} s^l (t/s)^m (u/s)^n \ln^j(s). \tag{4}$$

The logarithm factor, first introduced by Frankowski and Pekeris [3], was important to overcome the Kinoshita [2] accuracy of $10^{-6}$ Ha.

# 3   Formalisms

In this section we will in particular introduce SCRPA and detail the r-RPA approach we have followed.

## 3.1   Standard, renormalized and self-consistent RPA

The standard and also self-consistent RPA equations can be quite straightforwardly derived from the equations of motion (EOM) [33,34] of excitation creation operators $\hat{Q}_\lambda^\dagger$, defined by

$$\hat{Q}_\lambda^\dagger|\Phi_0\rangle = |\Phi_\lambda\rangle, \quad \text{with} \quad \hat{Q}_\lambda^\dagger = |\Phi_\lambda\rangle\langle\Phi_0|,$$

with $\Phi_\lambda$ the excited states, both singlets and triplets, and $\Phi_0$ the ground state,

$$\hat{H}|\Phi_\lambda\rangle = E_\lambda|\Phi_\lambda\rangle,$$

of the full Hamiltonian $\hat{H}$,

$$
\begin{aligned}
\hat{H} &= \hat{H}^0 + \hat{W} = \\
&= \sum_{k_1 k_2} \epsilon_{k_1 k_2}^0 \hat{c}_{k_1}^\dagger \hat{c}_{k_2} + \frac{1}{4}\sum_{k_1 k_2 k_3 k_4} \bar{v}_{k_1 k_2 k_3 k_4} \hat{c}_{k_1}^\dagger \hat{c}_{k_2}^\dagger \hat{c}_{k_4} \hat{c}_{k_3},
\end{aligned}
$$

where $\hat{H}^0 = \hat{T} + \hat{V}_{\text{ext}}$ is the noninteracting Hamiltonian and $\epsilon_{kk'}^0$ is its matrix elements with respect to a basis set $\phi_k(r)$ over which we also define the creation/annihilation operators $\hat{c}_k^\dagger/\hat{c}_k$, while

$$\bar{v}_{k_1 k_2 k_3 k_4} = \langle\phi_{k1}\phi_{k2}|v|\phi_{k3}\phi_{k4}\rangle - \langle\phi_{k1}\phi_{k2}|v|\phi_{k4}\phi_{k3}\rangle \tag{5}$$

are the antisymmetrized matrix elements of the many-body interaction $v$, in this work the Coulomb interaction $v(r, r') = 1/|r - r'|$. The Hermitian conjugated annihilation operators are subject to the killing condition on the ground state,

$$\hat{Q}_\lambda|\Phi_0\rangle = 0. \tag{6}$$

From the equation of motion obeyed by the $\hat{Q}_\lambda^\dagger$, we can derive the equation [33–35]

$$\langle\Phi_0|[\delta\hat{Q}, [\hat{H}, \hat{Q}_\lambda^\dagger]]|\Phi_0\rangle = \Omega_\lambda\langle\Phi_0|[\delta\hat{Q}, \hat{Q}_\lambda^\dagger]|\Phi_0\rangle, \tag{7}$$

where $\Omega_\lambda = E_\lambda - E_0$ are the excitation energies measured from the ground state, and $\delta\hat{Q}$ is an arbitrary variation of the operator $\hat{Q}_\lambda^\dagger$, associated to a generic state of the Hilbert space $|\Phi\rangle = \delta\hat{Q}^\dagger|\Phi_0\rangle$. For a variant of the derivation of Eq. (7), see Ref. [36] where the minimization of an energy weighted sum rule is employed.

So far everything is exact. We understand that $\hat{Q}_\lambda^\dagger$ is a complicated many body operator which may be considered as a superposition of one body, two-body, ..., N-body operators. We now restrict, as an approximation, the $\hat{Q}_\lambda^\dagger$ operators to be of the one-body form

$$\hat{Q}_\lambda^\dagger = \sum_{k_1 \neq k_2} \chi_{k_1 k_2}^\lambda \hat{c}_{k_1}^\dagger \hat{c}_{k_2},$$

with both $k_1$ and $k_2$ running over all indices, besides the diagonal configurations. We obtain the secular equation

$$\sum_{k_1' k_2'} \mathcal{S}_{k_1 k_2, k_1' k_2'} \chi_{k_1' k_2'}^\lambda = \Omega_\lambda \chi_{k_1 k_2}^\lambda, \tag{8}$$

with the matrix $\mathcal{S}$ defined as

$$\mathcal{S}_{k_1 k_2, k_1' k_2'} = \langle \Phi_0 | [\hat{c}_{k_1}^\dagger \hat{c}_{k_2}, [\hat{H}, \hat{c}_{k_1'}^\dagger \hat{c}_{k_2'}]] | \Phi_0 \rangle (n_{k_2'} - n_{k_1'})^{-1},$$

where $n_k \delta_{kk'}$ is the single-particle density matrix

$$\langle \Phi_0 | \hat{c}_k^\dagger \hat{c}_{k'} | \Phi_0 \rangle = \delta_{kk'} n_k,$$

supposed, for convenience, to be diagonal. (This is an approximation. It can be avoided without formal problems by using the canonical basis which diagonalizes the single-particle density matrix, but usually it does not add much to the accuracy of the solution.) By developing the double commutator we obtain

$$\begin{aligned}
\mathcal{S}_{k_1 k_2, k_1' k_2'} =& (\epsilon_{k_1} - \epsilon_{k_2}) \delta_{k_1 k_1'} \delta_{k_2 k_2'} + (n_{k_2} - n_{k_1}) \bar{v}_{k_1 k_2' k_2 k_1'} \\
& + \Big[ -\delta_{k_2 k_2'} \frac{1}{2} \sum_{j_1 j_2 j_3} \bar{v}_{k_1 j_1 j_2 j_3} C_{j_2 j_3 k_1' j_1} - \delta_{k_1 k_1'} \frac{1}{2} \sum_{j_1 j_2 j_3} \bar{v}_{j_1 j_2 k_2 j_3} C_{k_2' j_3 j_1 j_2} \\
& + \sum_{j_1 j_2} (\bar{v}_{k_1 j_1 k_1' j_2} C_{k_2' j_2 k_2 j_1} + \bar{v}_{k_2' j_1 k_2 j_2} C_{k_1 j_2 k_1' j_1}) \\
& - \frac{1}{2} \sum_{j_1 j_2} (\bar{v}_{k_1 k_2' j_1 j_2} C_{j_1 j_2 k_2 k_1'} + \bar{v}_{j_1 j_2 k_2 k_1'} C_{k_1 k_2' j_1 j_2}) \Big] (n_{k_2'} - n_{k_1'})^{-1},
\end{aligned}$$

in terms of the cumulant of the two-particle correlation functions

$$C_{k_1 k_2 k_3 k_4} = \langle \Phi_0 | \hat{c}_{k_3}^\dagger \hat{c}_{k_4}^\dagger \hat{c}_{k_2} \hat{c}_{k_1} | \Phi_0 \rangle - n_{k_1} n_{k_2} [\delta_{k_1 k_3} \delta_{k_2 k_4} - \delta_{k_2 k_3} \delta_{k_1 k_4}],$$

and of the single-particle energies $\epsilon_k$ and basis set $\phi_k$ eigensolutions of the equation

$$[H^0 + V^{\mathrm{MF}}] \phi_k = \epsilon_k \phi_k, \tag{9}$$

with the mean-field potential given by

$$V_{k_1 k_2}^{\mathrm{MF}} = \sum_k \bar{v}_{k_1 k k_2 k} n_k. \tag{10}$$

(Note that *a priori* in the mean-field basis $\phi_k$ neither the kinetic energy, nor the external potential are diagonal separately). The correlation functions $C$ contain only the fully connected terms of the two-body density matrix, i.e., the fully correlated part.

The correlation functions $C$ can be expressed by the RPA solution, and thus Eq. (8), with the expression for $\mathcal{S}$ above, constitute the full self-consistent RPA (SCRPA) equations. If we neglect in $\mathcal{S}$ all two-body correlation functions $C$, we obtain the renormalized RPA (r-RPA) approach. Replacing additionally the correlated $n_k$ by the uncorrelated integer Hartree-Fock occupation numbers, $n_h^{\mathrm{HF}} = 1$ for holes and $n_p^{\mathrm{HF}} = 0$ for particles, we re-obtain the standard RPA equations with the exchange term [33–35]

$$\begin{pmatrix} A & B \\ B^* & A^* \end{pmatrix} \begin{pmatrix} X^\lambda \\ Y^\lambda \end{pmatrix} = \Omega_\lambda \begin{pmatrix} 1 & 0 \\ 0 & -1 \end{pmatrix} \begin{pmatrix} X^\lambda \\ Y^\lambda \end{pmatrix}, \tag{11}$$

with

$$A_{ph,p'h'} = (\epsilon_p^{\mathrm{HF}} - \epsilon_h^{\mathrm{HF}})\delta_{pp'}\delta_{hh'} + (n_h^{\mathrm{HF}} - n_p^{\mathrm{HF}})\bar{v}_{ph'hp'},$$
$$B_{ph,p'h'} = (n_h^{\mathrm{HF}} - n_p^{\mathrm{HF}})\bar{v}_{pp'hh'}.$$

Indeed in this case the mean-field potential [Eq. (10)] is exactly the Hartree potential plus the exchange (Fock) operator, and Eq. (9) is the Hartree-Fock equation, so that $\epsilon_k$ and $\phi_k(r)$ are the Hartree-Fock energies and wave functions. So the $\mathcal{S}$ matrix contains HF energies $\epsilon_k$ along the diagonal, while the kernel reduces to the $\bar{v}$ terms.

In this work we went beyond standard RPA towards self-consistency, but did not pursue full SCRPA. The latter task remains for the future. We followed the r-RPA approach where in Eq. (11) all HF occupation numbers and energies are replaced by correlated ones, see, e.g., Catara *et al.* [16, 17]. In this approach, at each step of self-consistency a new, beyond Hartree-Fock, correlated mean-field electronic structure is calculated. The correlated electronic structure is characterized by noninteger occupation numbers $n_h$ and $n_p$, unlike the integer uncorrelated Hartree-Fock occupation numbers. The depletion/repletion with respect to HF uncorrelated occupation numbers can, e.g., be calculated from the correlated RPA amplitudes $\chi_{hp}^{\lambda}$ (*number operator* method [16, 33])

$$n_p = \frac{1}{2}\sum_{\lambda h}(n_h - n_p)|\chi_{hp}^{\lambda}|^2, \tag{12}$$

$$n_h = 1 - \frac{1}{2}\sum_{\lambda p}(n_h - n_p)|\chi_{hp}^{\lambda}|^2. \tag{13}$$

(The same result can be obtained with other formulations [37]). For small depletion/repletion one can replace the occupation numbers on the right-hand side with uncorrelated Hartree-Fock 0/1 occupation numbers. These expressions are correct to second order in $|\chi_{hp}^{\lambda}|$. Catara *et al.* [16, 17] considered higher-order corrections but we will see that in helium the depletion/repletion of occupation numbers constitute a correction of less than 1%, so that higher-order corrections are negligible, and stopping at second order is safe. Note that the occupation numbers of Eqs. (12) and (13) fulfill Luttinger's theorem, since the particle number $N$ is conserved:

$$\sum_h n_h + \sum_p n_p = N.$$

Also we will restrict the configuration space to particle-hole (hole-particle).

So, starting from standard RPA, after having solved the RPA equations and having calculated the $\chi$ amplitudes, we recalculate the correlated occupation numbers using Eqs. (12) and (13), the new mean-field potential using Eq. (10) and the new correlated energies $\epsilon_k$ using Eq. (9). The procedure is cycled till self-consistency. This r-RPA approach can be considered an approach towards SCRPA with the important simplification that the two-body correlation functions in the $\mathcal{S}$ matrix are neglected, but correlations are at least self-consistently introduced in the occupation numbers and in the single-particle energies that now depart from the uncorrelated HF expressions (see Fig. 1).

Finally, this methodology also allows one to calculate the total energy of the ground state, that is the correlation contribution of RPA, $E_c^{\mathrm{RPA}}$, or SCRPA, $E_c^{\mathrm{SCRPA}}$, to be added to the Hartree-Fock $E^{\mathrm{HF}}$ kinetic, external, Hartree and exchange contributions to the total energy, $E^{\mathrm{RPA}} = E^{\mathrm{HF}} + E_c^{\mathrm{RPA}}$. The correlation contribution can be calculated by [see Eq. (8.111) in Ref. [35]]

$$E_c = -\sum_{\lambda>0}\Omega_\lambda\sum_{ph}|\chi_{hp}^{\lambda}|^2,$$

but also by the expression [see Eq. (8.94b) in Ref. [35]]

$$E_c = \frac{1}{2} \sum_{\lambda > 0} \left( \Omega_\lambda^{\text{full}} - \Omega_\lambda^{\text{TDA}} \right),$$

(the sums over $\lambda$ run only over the positive $\Omega_\lambda$ energies) implying a difference between the excitation energies obtained by solving the full RPA Eq. (11), and excitation energies in the Tamm-Dancoff approximation (TDA), obtained by neglecting the coupling terms $B$ between the particle-hole and the hole-particle sectors of the full matrix in the solution of the Eq. (11). The two formulas gave the same results well within the accuracy quoted in this work, and so provided a cross check over the validity of the total-energy results. The same formulas were also used to calculate the total energy in the BSE approach.

To perform the renormalized RPA calculation on helium we first calculated the HF ground state and electronic structure energies and wave functions $\epsilon_i^{\text{HF}}, \phi_i^{\text{HF}}(r)$ by solving the Hartree-Fock equations

$$H_{\text{H}}(r)\phi_i^{\text{HF}}(r) + \int dr' \Sigma_x(r, r')\phi_i^{\text{HF}}(r') = \epsilon_i^{\text{HF}}\phi_i^{\text{HF}}(r),$$

where $H_{\text{H}}(r) = -\partial_r^2/2 + v_{\text{ext}}(r) + v_{\text{H}}(r)$ is the Hartree Hamiltonian and $\Sigma_x$ is the Fock exchange operator. We did not rely on pseudopotentials and rather use the full nuclear potential $v_{\text{ext}}(r) = -Z/r$ to reduce any source of inaccuracy in our comparison to the exact result. The HF calculation was carried out by the NWCHEM package [38]. With the HF electronic structure we calculated the $\mathcal{S}$ matrix of the RPA equation (8) and then solved it to get the standard RPA excitations (both singlet and triplet) energies $\Omega_\lambda$ and amplitudes $\chi^\lambda$. These are the excitations that we report in our tables and figures as (standard) RPA or TDHF, and are also the first-iteration result of an r-RPA calculation towards self-consistency. We then used the $\chi_{hp}^\lambda$ amplitudes to update the occupation numbers [Eqs. (12) and (13), where $\lambda$ run over both singlet and triplet excitations] and energies $\epsilon_k$ from Eq. (9), which are reinjected into the RPA equation to be solved again for new $\chi^\lambda$ amplitudes. The procedure was repeated until self-consistency, (at most four cycles were enough to achieve the $10^{-4}$ Ha accuracy we quote). The r-RPA calculations were carried out using a modified version of the FIESTA code [39, 40]. We used the *d-aug*-cc-pV5Z [41] correlation-consistent Gaussian basis set with angular momentum up to $l = 5$ and including a double set of diffuse orbitals. This was the most converged basis set and the best available to us.

## 3.2 QMC

We performed variational and diffusion quantum Monte Carlo (VMC and DMC) calculations [42, 43] of the nonrelativistic ground-state energy of an isolated all-electron helium atom with infinite nuclear mass. The CASINO code was used to perform our calculations [44]. The ground-state wave function is nodeless and hence the DMC algorithm is unbiased in the limit of zero time step, infinite walker population, and sufficiently long equilibration time.

We used a trial wave function of Slater-Jastrow form [43]:

$$\Psi(\mathbf{r}_1, \mathbf{r}_2) = \phi_{1s}^{\text{HF}}(r_1) \phi_{1s}^{\text{HF}}(r_2) \exp(J), \tag{14}$$

where the Jastrow exponent is [45]

$$
\begin{aligned}
J \;=\; & \sum_l \alpha_l r_{12}^l (r_{12}-L_u)^3 \Theta(L_u - r_{12}) \\
& + \sum_i \sum_m \beta_m r_i^m (r_i - L_\chi)^3 \Theta(L_\chi - r_i) \\
& + \sum_{l,m,n} \gamma_{lmn} r_1^l r_2^m r_{12}^n (r_1 - L_f)^3 (r_2 - L_f)^3 \\
& \hspace{3cm} \times \Theta(L_f - r_1)\Theta(L_f - r_2),
\end{aligned}
\tag{15}
$$

where $\Theta$ is the Heaviside function. The electron orbital $\phi_{1s}^{\mathrm{HF}}$ in the Slater part was calculated using Hartree-Fock theory and was represented numerically on a radial spline grid, allowing the electron-nucleus Kato cusp condition to be satisfied [46, 47]. The Jastrow exponent consisted of polynomial electron-electron, electron-nucleus, and electron-electron-nucleus terms, which were smoothly truncated at distances of $L_u = 8$, $L_\chi = 8$, and $L_f = 6$ Bohr, respectively [45]. Constraints were imposed on the parameters $\alpha_l$, $\beta_m$, and $\gamma_{lmn}$ to enforce the electron-electron Kato cusp condition and to avoid interfering with the electron-nucleus cusp condition; the remaining parameters were optimized. The Jastrow factors used in the great majority of QMC calculations are of this form or similar. Since the exact helium-atom wave function is a function solely of the electron-nucleus and electron-electron distances, the helium atom is a favorable case for our Jastrow exponent, which is a polynomial expansion in these distances. Free parameters in our trial wave function were optimized by energy minimization [48]. Our wave function contained 42 free parameters, and optimization of the wave function required about 32 core hours of computational effort. The resulting VMC energy is $-2.90372220(7)$ Ha. This is lower than the VMC energy [$-2.903693(1)$ Ha] reported with a similar form of Jastrow factor in Ref. [45] due to the use of a different optimization method.

In our DMC calculations we used time steps of 0.002 and 0.008 $\mathrm{Ha}^{-1}$, with corresponding target populations of 1024 and 256 walkers. The resulting DMC energies were extrapolated linearly to zero time step and hence, simultaneously, to infinite population. The resulting DMC energy of a helium atom with infinite nuclear mass is $-2.9037246(9)$ Ha. The total cost of the two DMC calculations was 121 core hours.

The DMC result is within error bars of the exact energy, as shown in Table 2. This is to be expected, as DMC is a statistically exact method for helium. However, a small difference between the VMC result and the exact result can be seen. This is due to the finite extent and order of the polynomials in the wave function, and the use of a finite number of random configurations in the wave-function optimization.

## 3.3 CI

The configuration interaction (CI) results we report here are standard calculations for which we invite the reader to refer to the specialized literature [49]. We were able to perform full-CI [49] calculations for all Gaussian basis sets employed here, except for the *d-aug*-cc-pV5Z which is fundamental to get the best convergence for excited states. This basis set is already too large to allow a full-CI calculation, at least for the computing resources available to us. This is already an important indication of the extent to which a given methodology, here CI, is able to achieve in practice. However, for *d-aug*-cc-pV5Z we performed an iterative-configuration expansion configuration interaction (ICE-CI) calculation [50, 51] and we checked, within the cc-pV5Z and the *d-aug*-cc-pVQZ basis sets, that ICE-CI provides results that are indistinguishable from full-CI, in particular on the ground state where the difference is $< 10^{-9}$ Ha, and remains generally at $10^{-9}$ Ha for most excited states, except in one case, where the difference was found to be $7 \cdot 10^{-6}$ Ha (see Table 1), well beyond what can be considered the accuracy

Table 1: Helium excitation energies in atomic units (Ha), comparing Full-CI and ICE-CI results within the cc-pV5Z and the *d-aug*-cc-pVQZ Gaussian basis sets. The zero of energy is the full ionization onset, $He^{++} + 2e^-$.

| | cc-pV5Z | | | *d-aug*-cc-pVQZ | |
|---|---|---|---|---|---|
| $n^S L$ | Full-CI | ICE-CI | $n^S L$ | Full-CI | ICE-CI |
| $1^1 S$ | −2.903151884 | −2.903151884 | $1^1 S$ | −2.902536607 | −2.902536607 |
| $2^3 S$ | −2.041940640 | −2.041940640 | $2^3 S$ | −2.174798591 | −2.174798592 |
| $2^1 S$ | −1.923273478 | −1.923273482 | $2^1 S$ | −2.145020288 | −2.145020287 |
| $2^3 P$ | −1.714041381 | −1.714041383 | $2^3 P$ | −2.130703422 | −2.130703422 |
| $2^1 P$ | −1.593255618 | −1.593255621 | $2^1 P$ | −2.119799159 | −2.119799159 |
| $3^1 S$ | −0.588140506 | −0.588140506 | $3^3 S$ | −2.063091342 | −2.063091342 |
| $3^3 S$ | −0.575726092 | −0.575726092 | $3^1 S$ | −2.046569475 | −2.046576198 |
| $3^3 P$ | −0.390104384 | −0.390104386 | $3^3 D$ | −1.920654679 | −1.920654679 |
| $3^1 P$ | −0.326412907 | −0.326412909 | $3^1 D$ | −1.920163475 | −1.920163475 |

of CI with respect to the exact solution. It would be unfair not to say that, beyond ICE-CI, there are several other methods able to provide near-full-CI energies, like CIPSI, CCSD, etc. We again invite the reader to refer to the specialized literature [49, 52–54] In the rest of the paper we therefore quote these results (with a number of decimal digits equal to or less than 6) as CI *tout court*, irrespective of whether they were obtained using full-CI or ICE-CI. All CI calculation were carried out using the publicly available ORCA code [51, 55].

### 3.4 HF+dRPA, *GW*+dRPA, BSE

The results and the details of many-body perturbation theory using the Bethe-Salpeter equation have been already reported by some of us in a previous publication [56] to which we refer the reader for both the theory and the parameters used in the calculation. In practice, the BSE equation is very similar to the standard RPA equation [Eq. (11)]. The major differences are that the RPA kernel $\bar{v}$, Eq. (5), is replaced by a kernel

$$K^{\text{BSE}}_{ijkl} = \langle ij|v|kl \rangle - \langle ij|W|lk \rangle,$$

in the BSE equation, where the second term has been replaced by matrix elements of the screened Coulomb interaction $W$, instead of the bare Coulomb interaction $v$. Another major difference is that the BSE calculation is done on top of an already correlated $GW$ electronic structure instead of the HF uncorrelated electronic structure used in standard RPA (see Fig. 1).

In this work we add some other new results obtained using the dRPA approximation, that is the direct RPA without exchange diagrams. The difference between the two can be understood when looking at the Feynman diagrams entering the irreducible polarizability $\tilde{\Pi}$ (see Fig. 1): in the dRPA polarizability only the particle-hole bubble (ring) diagram enters, while in the full RPA both the ring bubble and also the particle-hole exchange bubble diagram enter. In both cases then the irreducible polarizabilities $\tilde{\Pi}$ are resummed up to infinity to get the reducible polarizability $\Pi$ (last line of Fig. 1). The dRPA is therefore a lower level of approximation. Another often used name for the dRPA is *ring approximation*, with reference to the diagrams taken into account.

We report new results for helium obtained using this dRPA on top of both Hartree-Fock and also $GW$ electronic structures. So, the particle-hole ring bubbles are calculated using electron and hole Green's functions relying in one case on the HF electronic structure (HF+dRPA), and in the other on the quasiparticle $GW$ ($GW$+dRPA, see Fig. 1). The comparison between the

HF+dRPA with the RPA results and between the $GW$+dRPA with the BSE result, will show us the effect of the electron-hole interaction represented, in the first case by the bare Coulomb interaction, and by the screened Coulomb interaction in the second (Fig. 1). Like in all the other cases, we used the same ingredients, Gaussian basis sets (*d-aug*-cc-pV5Z), and parameters of the calculation so to allow the most faithful comparison between methods. The calculations were carried out using again the FIESTA code, switching off the electron-hole interaction term of the kernel.

### 3.5 TDDFT

TDDFT calculations share a lot of similarities with the standard RPA. In TDDFT excitation energies and amplitudes are calculated by solving also the RPA equations [Eq. (11)], which in chemistry are called the Casida equations [57, 58]. The differences with respect to standard RPA are that (see Fig. 1): 1) The DFT Kohn-Sham electronic structure is used instead of the HF electronic structure to calculate the zero-order polarizability; 2) The kernel $\bar{v}$, Eq. (5), of the standard RPA equations is replaced by a TDDFT kernel $f^{\text{TDDFT}}$ given by

$$f_{ijkl}^{\text{TDDFT}} = \langle ij|v|kl \rangle + \langle ij|f_{xc}|kl \rangle. \tag{16}$$

The first term is exactly the same in standard RPA, BSE and TDDFT kernels. TDDFT replaces the second exchange term of RPA or the $W$ term of BSE, by a direct term called exchange-correlation kernel, $f_{xc}$, which is defined as the functional derivative of the exchange-correlation potential with respect to density (the second functional derivative of the exchange-correlation energy):

$$f_{xc}[\rho](x_1, x_2) = \frac{\delta v_{xc}[\rho](x_1)}{\delta \rho(x_2)} = \frac{\delta^2 E_{xc}[\rho]}{\delta \rho(x_2)\delta \rho(x_1)}.$$

TDDFT is an in-principle-exact framework to calculate neutral excitation energies and oscillator strengths. However, the exact form of $f_{xc}$ is in general unknown. The latter in particular is in principle a dynamical quantity depending on time and hence on frequency [59]. So one must resort to approximations. The adiabatic local-density approximation (ALDA or TDLDA) is one of the most popular and consists of taking the functional derivative of the DFT local-density approximation to the exchange-correlation potential with respect to the density. Here we report calculations using this TDLDA approximation on top of both a DFT-LDA Kohn-Sham electronic structure as well as the exact DFT Kohn-Sham electronic structure. The latter is reported in Ref. [60, 61], and was done using a real-space-real-time code. On the other hand, we carried out DFT-LDA+TDLDA calculations using the NWCHEM code relying once again on the same basis set and calculation parameters as in all other calculations, in particular the *d-aug*-cc-pV5Z basis.

Finally, we also report the results of a DFT-LDA+dRPA calculation, that is using the dRPA approximation on top of a DFT LDA Kohn-Sham electronic structure. This is equivalent to a TDDFT calculation neglecting completely the exchange-correlation kernel, $f_{xc} = 0$, in Eq. (16) [62]. The comparison between DFT-LDA+dRPA and DFT-LDA+TDLDA results shows the effect of the approximated exchange-correlation kernel $f_{xc}$.

## 4 Results

In this section we will compare the results provided by the different methods, starting with the ground-state energy and then moving to excitations.

Table 2: Ground-state energy as calculated by different many-body approaches. The zero of energy is set to the full ionization onset, $He^{++} + 2e^-$. GGA refers to the PBE functional. The exact-DFT result quoted from Ref. [61] is calculated from the exact exchange-correlation potential obtained by reverse engineering from the exact Hylleraas solution. So, its accuracy only reflects the accuracy of the Hylleraas solution that must be known in advance, in contrast to the accuracies of all other methods which are genuine and real. We quote the exact-DFT result just to remind the reader of the scope of DFT, which should be the target of improved approximations.

| Method | Energy [Ha] |
| --- | --- |
| Noninteracting | −4 |
| Hartree | −1.9517 |
| HF | −2.8616 |
| DFT-LDA | −2.8348 |
| DFT-GGA | −2.8929 |
| Exact-DFT [61] | −2.903724377034118 |
| RPA (TDHF) | −2.9097 |
| r-RPA | −2.9085 |
| $GW$+BSE | −2.9080 |
| CI | −2.9032 |
| QMC-VMC (SJ) | −2.90372220(7) |
| QMC-DMC | −2.9037246(9) |
| **Exact** [10] | −2.90372437703411959831159245194404 |

## 4.1 Ground-state

In Table 2 we report the helium atom ground-state energy (in atomic units [Hartree] and setting the zero of the energies to the full ionization onset $He^{++} + 2e^-$) as calculated by all the methods we have considered. The exact result is quoted from the Hylleraas-like Schwartz calculation [10], which achieved an accuracy of 35 decimal digits, further confirmed by later work [11]. The noninteracting energy [Eq. (2)] of 4.0 Ha presents a large error of ∼ 1 Ha ∼ 30 eV from the exact result, showing how important are the interactions between electrons and how crude is the independent-particle approximation when calculating energies. The simplest many-body method, the Hartree-Fock (HF) theory, provides a total energy of −2.8616 (our Gaussian *d-aug*-cc-pV5Z HF calculation converged up to $10^{-4}$ Ha), already an important reduction of the error by almost two orders of magnitude down to ∼ 0.04 Ha ∼ 1 eV. This cannot at all be considered chemical accuracy that requires an error one order of magnitude less. Nevertheless, Hartree-Fock already provides a reasonable answer at least for the total energy of the system, and we will see also for the ionization potential. The difference between the exact and the Hartree-Fock energy,

$$E_c = E^{\text{Exact}} - E^{\text{HF}}, \tag{17}$$

is the more rigorous definition of the correlation energy. In helium, one of the few real systems where we know the exact total energy, we can calculate exactly the correlation energy and see that it is $E_c = -0.042 \, \text{Ha} = -1.15 \, \text{eV}$ (see Table 5), only 1.4% of the total energy.

Next we analyze the DFT-LDA result which presents an error larger (almost the double) than that of HF: 0.07 Ha ∼ 1.9 eV. A DFT generalized gradient approximation (GGA) [63,64] calculation (we used the most popular PBE functional [26]) reduces the error below the HF one: 0.01 Ha ∼ 0.3 eV. Notice that these are errors of the approximation, LDA or GGA (PBE),

Table 3: Ground-state correlation energy for the different many-body approaches, obtained by subtracting from the total ground-state energy (Table 2) the Hartree-Fock energy calculated at the same *d-aug*-cc-pV5Z Gaussian basis set (Table 2 second line). The exact correlation energy, calculated by Eq. (17), is converged to the same accuracy of the Hartree-Fock *d-aug*-cc-pV5Z calculation ($10^{-4}$ Ha.)

| Method | Correlation energy [Ha] |
|---|---|
| RPA (TDHF) | −0.0481 |
| r-RPA | −0.0469 |
| *GW*+BSE | −0.0464 |
| CI | −0.0416 |
| **Exact** | −0.0421 |

not of DFT, which is in principle an exact theory to calculate the total ground-state energy. The latter is just an idealistic statement which is better to avoid. Nevertheless, these statements are important to identify the scope and the limits of a theory, and orient the research to the real challenges within these limits [65]. Helium is one of the few cases where this statement is not purely idealistic, and the "exact DFT" that mathematical theorems guarantee to exist, can be really touched by hands. Thanks to the existence of the exact Hylleraas solution, the exact exchange-correlation potential of DFT can be calculated by reverse engineering [61,66]. The exact Hylleraas ground-state wave functions allows the calculation of the exact electron density, and from the latter we can calculate the only occupied DFT Kohn-Sham wave function. Knowing the exact Kohn-Sham occupied wave function and its corresponding Kohn-Sham energy (equal to the exact ionization potential also provided by the Hylleraas solution), the Kohn-Sham equation can be inverted to provide the exact exchange-correlation potential of DFT. This is the potential plotted in Fig. 8 of Ref. [61]. Using the exact exchange-correlation (XC) potential we can run the exact DFT and, for example, calculate the total ground-state energy (Table I of Ref. [61] reported in our Table 2) which, with no surprise, coincides with the exact Hylleraas energy. So for helium exact DFT is something more than only an idealistic theory. We cannot predict anything not already provided by the Hylleraas solution, but we can at least study the DFT methodology. Unlike all other entries in Table 2, the "Exact-DFT" line is there not to indicate the actual performances of DFT in general, but just to show that an exact exchange-correlation potential exists and is able to provide the exact ground-state total energy by a *mono-determinantal* (Kohn-Sham) approach (but not other quantities outside the scope of Kohn-Sham DFT), and it is thus meaningful to search for approximate functionals that try to be as close as possible to the exact potential also in the general case [65].

Next in our table we have a bunch of approximations that improve with respect to Hartree-Fock up to one order of magnitude ($\sim 0.004$ Ha $\sim 0.12$ eV for the *GW*+BSE ground-state

Table 4: CI ground-state energy calculation, Gaussian basis set convergence.

| CI | cc-pV$x$Z | *d-aug*-cc-pV$x$Z |
|---|---|---|
| TZ | −2.900232 | −2.900608 |
| QZ | −2.902411 | −2.902537 |
| 5Z | −2.903152 | −2.903202 |
| Extrapolation | −2.903878 | −2.903840 |
| **Exact** [9] | | **−2.903724** |

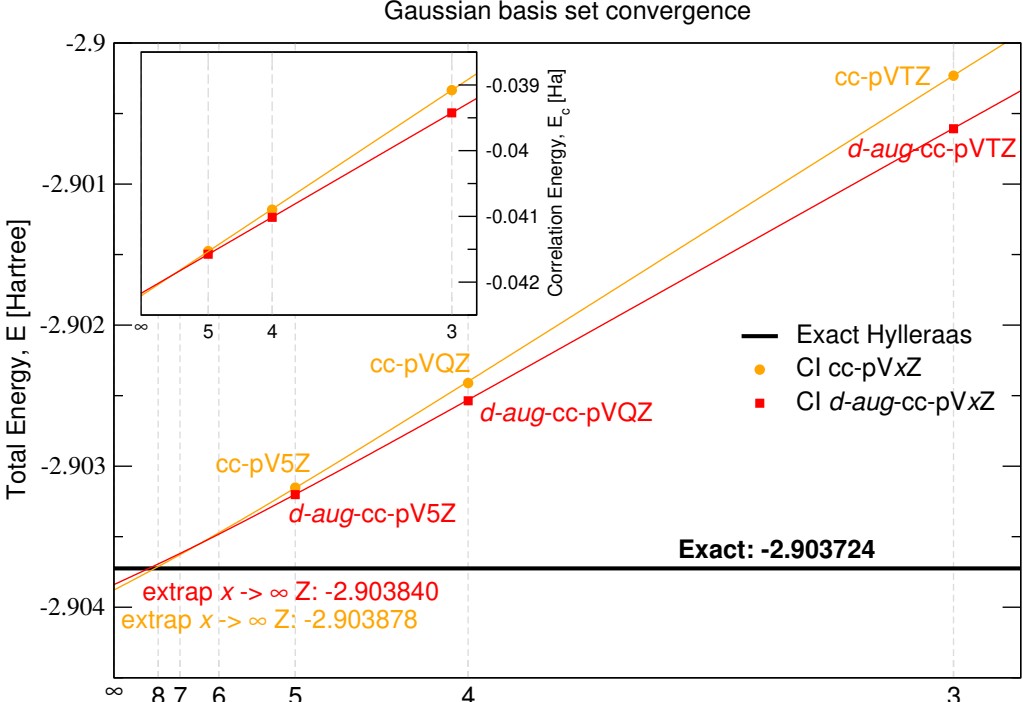

Figure 3: He atom ground-state $1^1S$ energy by CI calculations using increasing Gaussian basis sets. In orange: calculations using the standard cc-pV$x$Z Gaussian basis set at increasing $x$ (orange circles) and their extrapolation to $x \rightarrow \infty$ (orange line). Red: calculations using the double augmented $d$-$aug$-cc-pV$x$Z Gaussian basis set at increasing $x$ (red squares) and their extrapolation to $x \rightarrow \infty$ (red line). The Hartree-Fock and the correlation energies were separately fit to different formulas: the exponential function $E^{\mathrm{HF}}(x) = E^{\mathrm{HF}}_\infty + ae^{-bx}$ for Hartree-Fock, and a power law $E^c(x) = E^c_\infty + cx^{-3}$ for correlation (see inset). Black line: Exact Hylleraas-like calculation ground-state energy [10]. The zero of energy is set to the full ionization onset, $\mathrm{He}^{++} + 2e^-$.

energy). The standard RPA (TDHF) result presents a consistent improvement with respect to HF. Then both our r-RPA and the $GW$+BSE result improve almost by the same, non-negligible but small amount, with respect to standard RPA.

The first result that starts to be within the level of chemical accuracy (usually set to 1 kcal/mol $\sim 0.0016$ Ha $\sim 0.043$ eV [49]) is the CI result. In Table 2 we quote our best converged $d$-$aug$-cc-pV5Z result, presenting an error with respect to the exact result of $5 \cdot 10^{-4}$ Ha. However, looking to Table 4 we can see that quadruple-Z Gaussian basis sets are at the limits of chemical accuracy, and triple-Z, often the only possibility for molecular calculations and by many considered as the golden standard, are well outside. Our study here tried also to investigate to what extent the accuracy of CI can be improved. Helium is a very favorable case also for CI since the presence of only two electrons limits the configurations to be taken in consideration to singles and doubles only, with no need to include triples and beyond. Nevertheless a CI calculation is to be done within a basis, here as in most chemical calculations using a finite, incomplete Gaussian basis set. This limits the accuracy of the calculation due to two factors: 1) the number of configurations taken into account is limited by the number of elements in the basis set; for example, in the cc-pV5Z we have 55 basis elements, so that we

can at best take into account all the singles and doubles configurations out of 55 Hartree-Fock orbitals. 2) the chosen, localized or delocalized, basis set can limit the representation of the exact wave functions; for example, it can be very hard to represent the highest, almost free, excited states by using a necessarily limited set of localized Gaussians. We tried to study to which extent the accuracy of CI with respect to the previous issues can be pushed by applying a standard [67–69] extrapolation technique over the $x$-tuple zeta Gaussian basis set series, towards the limit $x \to \infty$. We fit the Hartree-Fock total energies calculated at both the cc-pV$x$Z and the $d$-$aug$-cc-pV$x$Z basis sets to the exponential function $E^{HF}(x) = E^{HF}_\infty + a e^{-bx}$, and separately the correlation energies to the power law $E^c(x) = E^c_\infty + c x^{-3}$. Figure 3 is a plot of this extrapolation technique compared to the exact Hylleraas energy (in the inset the $x^{-3}$ linear extrapolation for the correlation energy only). We report in Table 4 the extrapolated values, $E^{CI}_\infty = E^{HF}_\infty + E^c_\infty$. It can be seen that the extrapolation overshoots the exact result. With respect to the 5Z basis it provides a reduction of the error by a factor 5, but it is unable to go below an error of $10^{-4}$ Ha. The 6Z basis would present an error of the same magnitude of the extrapolated values, so it would not be convenient to go for the extrapolation once at the level, say, of 7Z or 8Z. (Of course these values might be system dependent.) Our analysis seems to show that a Gaussian CI extrapolation technique towards the exact result is improved by the *augmentation* of the basis set. This implies that delocalized basis set elements, that we will see are fundamental for the description of excited states, are also important for an accurate description of the ground-state wave function and energy.

Quantum Monte Carlo is the most accurate among the many-body methods studied here. The VMC approach generally relies on a Slater-Jastrow *Ansatz* for the variational trial wave function, and this form is used in nearly all QMC codes. Our VMC calculation achieves a random error of only $7 \cdot 10^{-8}$ Ha, with the systematic error (bias due to the restricted form of the trial wave function and the method used to optimize the free parameters) being $2.18(7) \cdot 10^{-6}$ Ha. Our DMC calculations using this VMC-optimized Slater-Jastrow wave function achieved a statistical error of $9 \cdot 10^{-7}$ Ha, with no evidence of systematic bias. This demonstrates that QMC is effectively able to achieve the experimental accuracy of Herzberg [70] used in the hystorical theory-experiment comparison of Pekeris [71]. However, as noted earlier, helium is a very favorable case for QMC because the ground-state wave function is nodeless; hence fixed-node DMC is unbiased, i.e., one can obtain arbitrarily precise and accurate DMC results by running for times as long as necessary. The present DMC calculation lasted 121 core hours. The statistical error bar falls off as the reciprocal of the square root of the computational effort. So the error bar can easily be reduced further, but the level of precision achieved in Hylleraas calculations is completely unachievable with QMC in practice, or would require a significantly better trial wave function, together with an adaptation of a QMC code for high-precision work.

The VMC accuracy achieved is 6 significant decimal digits, similar to the accuracy of Kinoshita's 1957 Hylleraas-like calculation [2]. Fundamental to achieving this accuracy is the cusp-like exponential factor $e^{-ks}$ present in the Kinoshita wave function of Eq. (3). This factor is also present in the standard VMC SJ *Ansatz*. So standard VMC can achieve Kinoshita's, but not Frankowski and Pekeris's [3] accuracy of 14 significant decimal digits. Achieving the latter might require the logarithmic factor $\ln(s)$ of Eq. (4), which is absent in the standard VMC SJ *Ansatz* of multipurpose codes. One can easily implement such logarithmic behavior in the VMC *Ansatz*. Knowing from the literature the best Hylleraas result and the associated wave function, one could code an *Ansatz* modeled on the latter and possibly achieve the same accuracy within VMC. However, this is not the criterion we have chosen in Table 2, where the results are deliberately obtained using "standard" methodology. In any case, the genuine QMC accuracies, compared to the rest of the methods that do not require advance knowledge of the exact solution, are already very impressive.

Table 5: Different contributions to the ground-state energy in atomic units (Hartree), as calculated in our Gaussian *d-aug*-cc-pV5Z HF calculation converged only up to $10^{-4}$ Ha (second column), and in the Exact-DFT calculation of Umrigar and Gonze [61] accurate to the quoted digits (third column). The 8th line reports the correlation energy, whose more rigorous definition is the difference between the exact and the Hartree-Fock energy. The zero of energy is set to the full ionization onset, $He^{++}+2e^-$.

| Energy contribution | HF | Exact-DFT | VMC |
|---|---|---|---|
| Kinetic | +2.8615 | +2.867082 | +2.90377(6) |
| External (*e-N*) | −6.7489 | −6.753267 | −6.75332(6) |
| Hartree | +2.0515 | +2.049137 | |
| Exchange | −1.0257 | | |
| Exchange-Correlation | | −1.066676 | |
| Many-body (*e-e*) | | | +0.94585(5) |
| Total | −2.8616 | −2.903724 | −2.90372220(7) |
| Correlation | −0.0421 | | |
| **Exact** [9] | **−2.903724377** | | |

## 4.2 Ground-state energy components

It is also instructive to analyze the individual components of the total ground-state energy. In Table 5 we report them for HF, QMC and once again also for exact DFT for illustration purposes rather than quantifying errors. The total energy benefits from the zero-variance principle (its error bar goes to zero as the trial wave function is optimized) in both VMC and DMC [43]. Hence the total energy is much more precisely and a little more accurately determined than its individual components. It is possible to calculate the exact energy components once the exact many-body wave function is available from a Hylleraas calculation. However we could not find them in the literature. We could anyway reconstruct what should be the exact components, for example from the virial theorem which in the case of Coulomb interacting systems says that the exact kinetic energy must be minus the total energy. By this argument we can see that only VMC provides the expected behavior for the kinetic energy, to within the error bars. This is not the case for HF: although the HF kinetic energy is virial with respect to its full HF total energy, it does not coincide with the exact kinetic energy. The HF kinetic energy is the average value of the kinetic energy operator over the HF single Slater determinant ground-state wave function, and the latter is just an approximation to the exact many-body ground-state wave function.

The same holds for exact DFT: the exact KS kinetic energy has nothing to do with the exact kinetic energy $T$. It is the kinetic energy $T_s$ of the fictitious Kohn-Sham independent-particle system, i.e. the sum of the average kinetic energies of the Kohn-Sham fictitious electrons. In fact, the difference between the exact and the Kohn-Sham kinetic energies, $T-T_s$, is included in the DFT exchange-correlation energy $E_{xc}$ which, hence, contains also a part of the real kinetic energy. Here we can evaluate this part to be +0.036642 Ha: this is almost the same magnitude (with change of sign) as the correlation energy rigorously defined by Eq. (17), $E_c = -0.0421$ (8th line in Table 5). So, this kinetic contribution to the defined total exchange-correlation energy $E_{xc}$ of DFT is not negligible at all with respect to the correlation contribution.

The electron-nucleus external energy can be calculated once again exactly (within the error bar) by VMC as the average of the external potential local operator over the VMC wave function. The external energy can also be in principle calculated exactly within DFT: to calculate this quantity the full many-body wave function is not needed, just the electronic density, which

Table 6: He excitation energies in Hartree and eV, comparison between BSE and CI calculated at the *d-aug*-cc-pV5Z basis against the exact [32] result. The zero of energy is set to the He ground state $1^1S$.

| $n^S L$ | BSE | CI | **Exact** | $n^S L$ | BSE | CI | **Exact** |
|---|---|---|---|---|---|---|---|
| | atomic units [Ha] | | | electronvolt [eV] | | | |
| $2^3S$ | 0.7271 | 0.7282 | **0.7285** | $2^3S$ | 19.786 | 19.815 | **19.824** |
| $2^1S$ | 0.7676 | 0.7577 | **0.7578** | $2^1S$ | 20.888 | 20.618 | **20.621** |
| $2^3P$ | 0.7724 | 0.7714 | **0.7706** | $2^3P$ | 21.018 | 20.991 | **20.969** |
| $2^1P$ | 0.7894 | 0.7818 | **0.7799** | $2^1P$ | 21.480 | 21.274 | **21.222** |
| $3^3S$ | 0.8427 | 0.8404 | **0.8350** | $3^3S$ | 22.930 | 22.868 | **22.722** |
| $3^1S$ | 0.8637 | 0.8565 | **0.8425** | $3^1S$ | 23.502 | 23.307 | **22.926** |
| $3^3P$ | 0.9514 | 0.9542 | **0.8456** | $3^3P$ | 25.890 | 25.965 | **23.010** |
| $3^3D$ | 0.9645 | 0.9617 | **0.8481** | $3^3D$ | 26.247 | 26.169 | **23.078** |
| $3^1D$ | 0.9663 | 0.9621 | **0.8481** | $3^1D$ | 26.294 | 26.180 | **23.078** |
| $3^1P$ | 0.9928 | 0.9829 | **0.8486** | $3^1P$ | 27.015 | 26.746 | **23.092** |

is provided exactly with the exact DFT. The external energies of exact DFT and VMC coincide. This of course is not the case in approximate (LDA, GGA, etc.) DFT. On the other hand, HF provides only an approximate electronic density, and so the external electron-nucleus energy provided by HF is only an approximation for this component.

Ambiguities related to the definitions start to arise when looking at the Hartree energy. At the beginning this component was defined with respect to a particular method, the Hartree or the Hartree-Fock method. These two methods already provide a different estimate for the Hartree energy, due to the fact that the ground-state wave functions and electronic densities are different. However the Hartree energy can be defined as the classical component to the many-body electron-electron energy:

$$E_{\mathrm{H}} = \int d^3r\, d^3r' \frac{\rho(\boldsymbol{r})\rho(\boldsymbol{r'})}{|\boldsymbol{r} - \boldsymbol{r'}|}.$$

With this definition, one can see that exact DFT also provides the exact Hartree energy, again because the density is exact. And we can measure the error in this component within HF theory, which is related to the error in the HF external energy. In principle the charge density and hence Hartree energy can be calculated within QMC, but in practice QMC directly evaluates the many-body electron-electron interaction energy (6th line of Table 5). Likewise for the exchange and correlation energies. In fact, the exchange energy can only be defined once the exchange operator, which relies on single-particle wave functions, is defined, what is meaningless in a QMC framework. A meaningful exchange energy can only be defined within the Hartree-Fock method and not within DFT even in the exact case. An exchange energy defined using the same HF shape for the exchange operator but using Kohn-Sham (KS) wave functions, i.e. the wave functions of the noninteracting KS electrons, has not the same physical interpretation as the genuine Hartree-Fock exchange. In DFT normally one simply requires a full exchange-correlation functional/potential that takes into account all missing components together, including the kinetic energy not accounted for by the Kohn-Sham kinetic energy $T_s$. This is indeed the case for exact DFT where the exchange-correlation energy exactly provides the missing contribution (exchange plus correlation plus residual non-Kohn-Sham kinetic energy) to achieve the exact total ground-state energy (5th line in Table 5). Of course DFT LDA, GGA, or other approximations, should be evaluated for their error strictly done on this quantity or on the density [65].

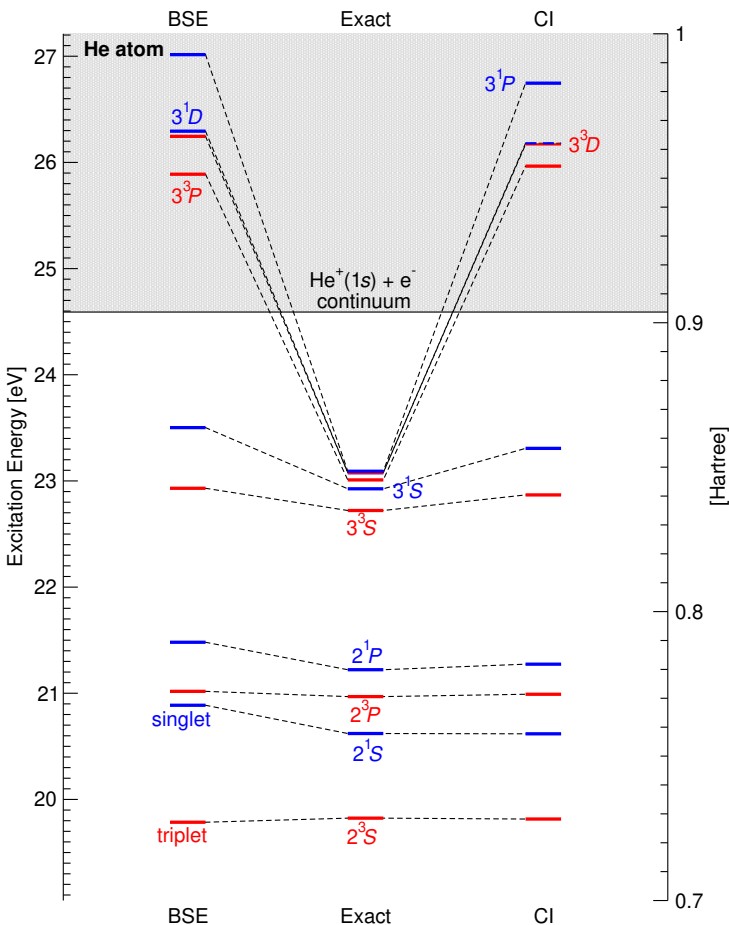

Figure 4: He excitation energies in Hartree and eV, comparing BSE and CI results calculated within the _d-aug_-cc-pV5Z basis against the exact [32] result. The zero of energy is set to the He ground state $1^1S$.

Finally from this table one can read off the exact value of the correlation energy, rigorously defined with respect to the total exact and Hartree-Fock energies by Eq. (17), and so have an estimate of its size and the only nonarbitrary and reliable evaluation of how strongly or weakly correlated a many-body system is. By comparing the correlation energy with the other contributions to the total energy, one can appreciate how important correlations are in a given system, whether correlations are going to change qualitatively the picture or they are only a quantitative adjustment. In helium the correlation energy is more than one order of magnitude less than all other components, only a small fraction $< 5\%$ of them, no matter how the other components are decomposed. So, the _helium atom can be classified as a weakly correlated system_.

## 4.3 Excitations

We now start to analyze excited states, starting from the comparison of CI and exact results (Table 6). The first three CI excitations are still within chemical accuracy from the exact result. The agreement is still acceptable, within 0.5 eV, for the next three excitations. However, starting from $3^3P$ the error jumps to 3 eV and more. These states are also provided as unbound since the ionization potential is set to 0.9037 Ha. This degradation is evidently a finite-basis effect. The lowest excited states are more localized and require few Gaussians to be represented accurately. Higher states get more and more delocalized and, consequently, require larger

Table 7: He excitation energies in atomic units (Hartree, top) and electronvolt (eV, bottom), comparison between different methods. The zero of energy is set to the He ground state $1^1S$. The exact DFT + TDLDA result is taken from Ref. [28] and it is the only one in the table calculated without using the Gaussian basis set. The last line reports the ionization potential (IP), i.e. the first ionization onset $He^+(1s) + e^-$, obtained from the last-occupied energy, $IP = -\epsilon_{1s}$, of the (depending on the methodology) HF, $GW$, DFT LDA or exact electronic structures. Notice that for DFT LDA based calculations we could have used the $IP_{\Delta SCF}^{DFT-LDA} = 0.8931$ Ha calculated by the $\Delta$SCF method, providing a fully bound Rydberg series, although severely red-shifted.

| $n^SL$ | Exact | CI | $GW$ + BSE | TDHF (RPA) | HF + dRPA | $GW$ + dRPA | DFT-LDA + dRPA | DFT-LDA + TDLDA | Exact-DFT + TDLDA |
|---|---|---|---|---|---|---|---|---|---|
| | | | | atomic units [Ha] | | | | | |
| $2^3S$ | **0.7285** | 0.7282 | 0.7271 | 0.7237 | 0.9396 | 0.9289 | 0.5826 | 0.5792 | 0.7351 |
| $2^1S$ | **0.7578** | 0.7577 | 0.7676 | 0.7759 | 0.9414 | 0.9307 | 0.5882 | 0.5853 | 0.7678 |
| $2^3P$ | **0.7706** | 0.7714 | 0.7724 | 0.7806 | 1.0136 | 1.0020 | 0.6381 | 0.6337 | 0.7698 |
| $2^1P$ | **0.7799** | 0.7818 | 0.7894 | 0.7997 | 1.0157 | 1.0041 | 0.6437 | 0.6340 | 0.7764 |
| $3^3S$ | **0.8350** | 0.8404 | 0.8427 | 0.8499 | 1.0574 | 1.0444 | 0.6693 | 0.6575 | 0.8368 |
| $3^1S$ | **0.8425** | 0.8565 | 0.8637 | 0.8732 | 1.0774 | 1.0644 | 0.7002 | 0.6872 | 0.8461 |
| IP | **0.9037** | 0.9179 | 0.9075 | 0.9179 | 0.9179 | 0.9075 | 0.5704 | 0.5704 | 0.9037 |
| | | | | electronvolt [eV] | | | | | |
| $2^3S$ | **19.824** | 19.815 | 19.786 | 19.692 | 25.569 | 25.276 | 15.853 | 15.760 | 20.003 |
| $2^1S$ | **20.621** | 20.618 | 20.888 | 21.115 | 25.618 | 25.324 | 16.007 | 15.928 | 20.893 |
| $2^3P$ | **20.969** | 20.991 | 21.018 | 21.242 | 27.581 | 27.266 | 17.363 | 17.244 | 20.947 |
| $2^1P$ | **21.222** | 21.274 | 21.480 | 21.762 | 27.639 | 27.323 | 17.515 | 17.251 | 21.127 |
| $3^3S$ | **22.722** | 22.868 | 22.930 | 23.128 | 28.773 | 28.421 | 18.214 | 17.891 | 22.770 |
| $3^1S$ | **22.926** | 23.307 | 23.502 | 23.762 | 29.317 | 28.963 | 19.054 | 18.701 | 23.024 |
| IP | **24.591** | 24.979 | 24.696 | 24.979 | 24.979 | 24.696 | 15.522 | 15.522 | 24.591 |

(more diffuse) basis sets. In particular we have found it is essential to use augmented Gaussian basis set to describe even the lowest excited states. Looking at Table 1 one can see that the cc-pV5Z basis presents an error of more than 3 eV already on the first excited state $2^3S$. This problem could be mitigated in large molecules because of the effect of basis elements sharing, i.e. the fact that each atom profits from the basis functions on its many neighbors. However, states towards the continuum of hydrogenic $He^+(1s)$ plus a free electron would require better adapted bases, e.g., plane waves. The states quoted in Table 6 (see also Table 1 for reference of convergence) are the only ones that could be unambiguously identified, though already in the unbound part of the spectrum.

The BSE approach is at the limit of chemical accuracy only for the first excited state $2^3S$ and generally presents a larger error than CI. Very importantly, we observe the same trend as in CI, with the characteristic breakdown at the level of the $3^3P$ excitation. From that point on we observe a large error of both CI and BSE, but the two methods are close to each other. The worsening is evidently due to basis-set incompleteness in both methods. The CI error can be regarded as mostly due to the incompleteness of the basis-set. With this assumption, we can evaluate the error due to the approximations done in the BSE formalism, independently from the basis set incompleteness error, by comparing directly BSE and CI results at the same basis set. We see that this BSE formalism error is no more than 0.2 eV, an error that allows us to describe the main physics of a system.

In Ref. [56] some of us already analyzed the results of $GW$+BSE in comparison to RPA

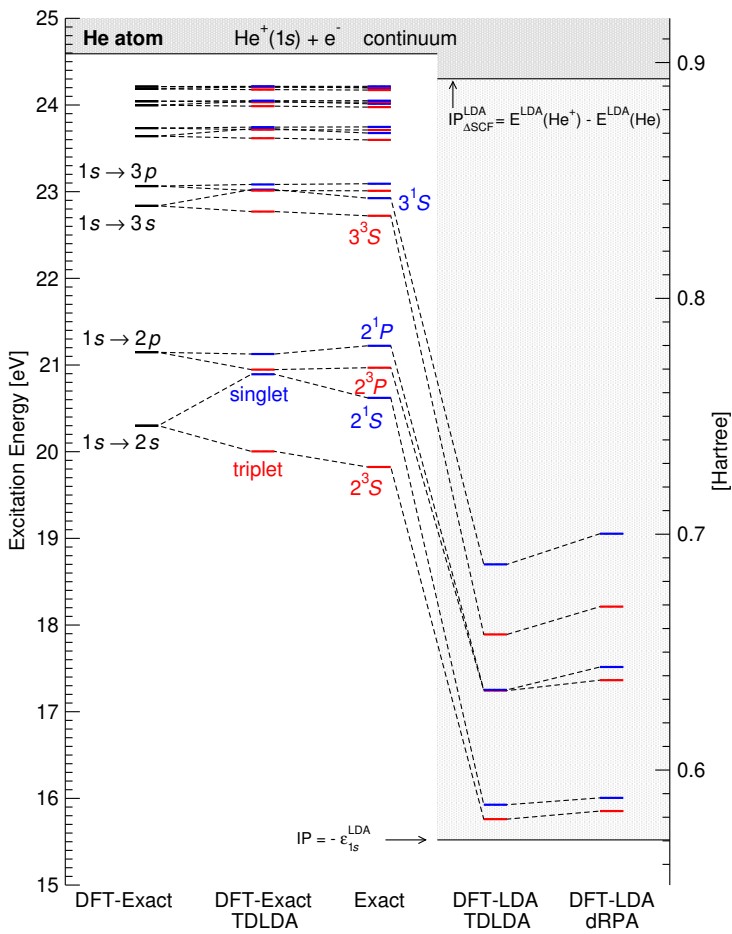

Figure 5: He excitation energies in Hartree and eV. The zero of energy is set to the He ground state $1^1S$. From the left: exact-DFT [60, 74]; TDLDA on top of exact-DFT [28]; exact spectrum [32]; TDLDA on top of DFT-LDA; dRPA on top of DFT-LDA. The DFT-LDA spectra are calculated at the *d-aug*-cc-pV5Z basis, while the exact-DFT are real-space calculations. Notice that for DFT LDA based calculations we have used as onset of the continuum the $IP^{DFT-LDA} = -\epsilon_{1s}^{DFT-LDA} = 0.5704$ Ha, but we could have better used the $IP_{\Delta SCF}^{DFT-LDA} = 0.8931$ Ha calculated by the $\Delta$SCF method. In the latter case we would have found a fully bound Rydberg series, and even severely red-shifted and overbound.

(TDHF). We now analyze the results one can obtain from a dRPA (ring) approximation on top of the HF or the *GW* quasiparticle electronic structure. It can be seen in Table 7 that the excitation energy is strongly overestimated in both approaches, a nonrigid shift of 5–7 eV, and a slightly larger one with HF+dRPA. The difference between *GW*+BSE and *GW*+dRPA is a term that introduces electron-hole (excitonic) screened interaction effects. This also holds for RPA (TDHF) and HF+dRPA, with the difference that we start from uncorrelated Hartree-Fock energies and the electron-hole interaction is unscreened. One can see that this electron-hole interaction term is very important at least in this isolated system, like it has been found to be important in large band-gap insulators [72] and in molecules [40, 73].

In helium we know that the distance between the first ionization level and the full ionization is exactly 2 Ha. This is trivially given by the solution of the Schrödinger equation for the system He$^+$, which is a one-electron hydrogenic atom with $Z = 2$. So, the ionization potential could be obtained by subtracting this value of 2 Ha from the ground-state energy. However, in general, for systems with more than two electrons, this information is not available. We can

Table 8: Helium atom first dipole-allowed $^1S \to 2^1P$ transition oscillator strength $f_{1^1S \to 2^1P}$.

| | Exact | GW + BSE | TDHF (RPA) | HF + dRPA | GW + dRPA | DFT-LDA + dRPA | DFT-LDA + TDLDA |
|---|---|---|---|---|---|---|---|
| $f_{1^1S \to 2^1P}$ | **0.27616** | 0.2763 | 0.2916 | 0.1011 | 0.0996 | 0.1476 | 0.1848 |

then use Koopmans' theorem: the last occupied HF eigenvalue, and more so the corresponding GW quasiparticle energy, can be interpreted as minus the ionization potential (IP) of the atom. This is $IP^{HF} = 0.9179$ Ha for HF and $IP^{GW} = 0.9075$ for GW, against the exact $IP^{exact} = 0.9037$ Ha. Referring to these values, one can conclude that the excitation spectra of both HF+dRPA and GW+dRPA are unbound (even the first excitation lie above the IP), in contrast to the exact excitation spectrum, which presents a whole Rydberg series below the ionization onset. From this point one can see the importance of running a BSE (or a TDHF / RPA) calculation using a kernel that contains the electron-hole interaction (exchange term) beyond the direct term of the simpler dRPA.

In DFT the last occupied Kohn-Sham eigenvalue is the only one that can be physically interpreted as minus the ionization potential, that is the energy to strip an electron from the system [75–77]. In exact DFT the last eigenvalue exactly coincides with minus the ionization potential. In Ref. [61] that value was taken from the exact Hylleraas calculation and imposed for the inversion of the Kohn-Sham equation. In approximate DFT-LDA the last occupied Kohn-Sham eigenvalue is supposed to give an approximate ionization potential in order to estimate where the onset of the continuum of excitation occurs. This gives us $IP^{DFT-LDA} = 0.5704$ Ha. With respect to this value it turns out that the DFT-LDA + dRPA spectrum is also fully unbound (see Table 7 and Fig. 5). The same for a DFT-LDA + TDLDA spectrum. There is no Rydberg series before the onset of the continuum in DFT-LDA both with dRPA and TDLDA. Notice that if we had used the information that the hydrogenic 1-electron helium ground-state energy is exactly 2 Ha and calculated the IP as the 2-electron helium DFT-LDA ground-state energy (from Table 2) minus these 2 Ha, getting IP = 0.83 Ha, then we would have found a bound Rydberg series, although severely red-shifted. An always available and more convenient choice of the ionization potential could have been obtained by taking the difference between the DFT-LDA ground-state energies of the 2-electron and the 1-electron atoms, what is called the ΔSCF method [78]. Even though the 1-electron DFT LDA calculation is the most critically affected by the self-interaction problem and error (that anyway our 1-electron DFT LDA calculation quantified to just only 0.06 Ha), by cancellation of errors with the 2-electron calculation a better result can be obtained: $IP^{DFT-LDA}_{\Delta SCF} = 0.8931$ Ha. So, we argue that in atoms, finding or not an unbound Rydberg series in TDLDA (or dRPA) on top of DFT LDA calculations depends, to a large extent, on the choice of how the IP has been calculated.

The use of the exact DFT Kohn-Sham spectrum [60, 74] (Fig. 5 left side), for which the last occupied Kohn-Sham energy provides the exact ionization potential [75–77], allows us to recover a bound Rydberg series in good agreement with the exact result. Indeed, an approximate TDLDA calculation done on top of exact DFT [28] is not any more affected by the two drawbacks of the TDLDA calculation done on top of approximate DFT LDA, i.e. both the unboundness of the entire spectrum due to the misplaced ionization potential, and also the 3∼5 eV severe shift of all excitations measured with respect to the ground-state energy (see Fig. 5 and Table 7). Notice that, as is commonly done in solids, one can simulate this correction by applying a scissor operator to the DFT-LDA KS eigenvalue spectrum. The LDA KS HOMO-LUMO gap of 15.853 eV has to be brought not to the exact HOMO-LUMO gap = IP - EA

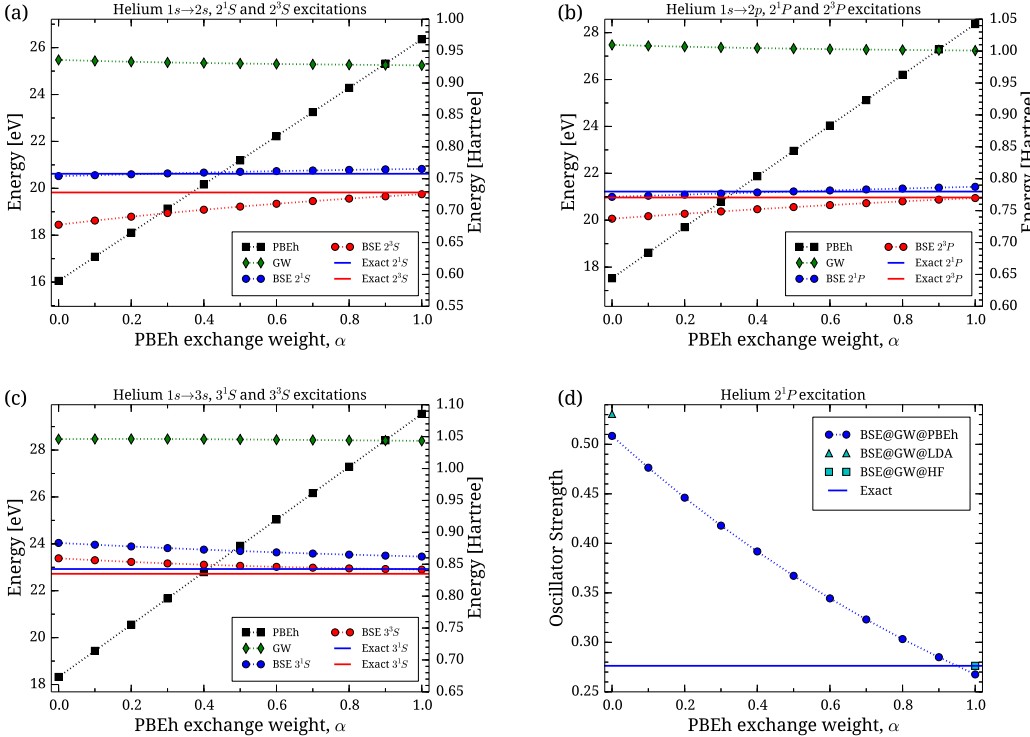

Figure 6: Starting point dependence, with respect to the PBEh exchange weight $\alpha$, of *GW* and BSE results. $\alpha = 0$ coincides with the original PBE [26] functional, while $\alpha = 1$ represents a full HF exchange plus the correlation contained in the PBE functional. a-c) The *GW* (green diamonds) and PBEh (black squares) gaps and the singlet (blue circles) and triplet (red circles) BSE excitation energies for a) $1s \to 2s$, b) $1s \to 2p$, and c) $1s \to 3s$, respectively. The corresponding exact singlet (blue solid line) and triplet (red solid line) excitation energies are reported from Ref. [32]. d) The $1^1S \to 2^1P$ BSE transition oscillator strength (blue circles). We report also the BSE transition oscillator strength obtained starting from pure HF (cyan square) and starting from pure DFT LDA (cyan triangle). The exact $1^1S \to 2^1P$ transition oscillator strength (solid line) is reported from Ref. [32].

(electron affinity) of $\sim 25$ eV, but rather to the exact "optical gap", i.e. at the 19.8 eV of the first excitation $2^3S$, or better at an average level of 20.2 eV between the singlet and triplet $2S$ excitations. A scissor operator rigid shift of 4–$\sim 4.4$ eV would better situate the DFT-LDA+TDLDA excitation spectrum.

To conclude this section we analyze the excitation oscillator strengths (Table 8). This is a quantity directly related to the quality of the wave functions. By checking oscillator strengths the different methodologies are evaluated with respect to the quality of the wave functions, independently from energies. We note the good performances of BSE, but also of RPA, against the unsatisfactory results of *GW*+dRPA and of HF+dRPA. Like for the excitation energies, both the unscreened kernel of RPA and the screened one of BSE are fundamental to achieve good oscillator strengths.

## 4.4 *GW* and BSE starting point dependence

All the previously quoted results with *GW* and BSE have been calculated starting from Hartree-Fock. This is the approach of the origins [19, 22, 23] and it also looks to us more significative

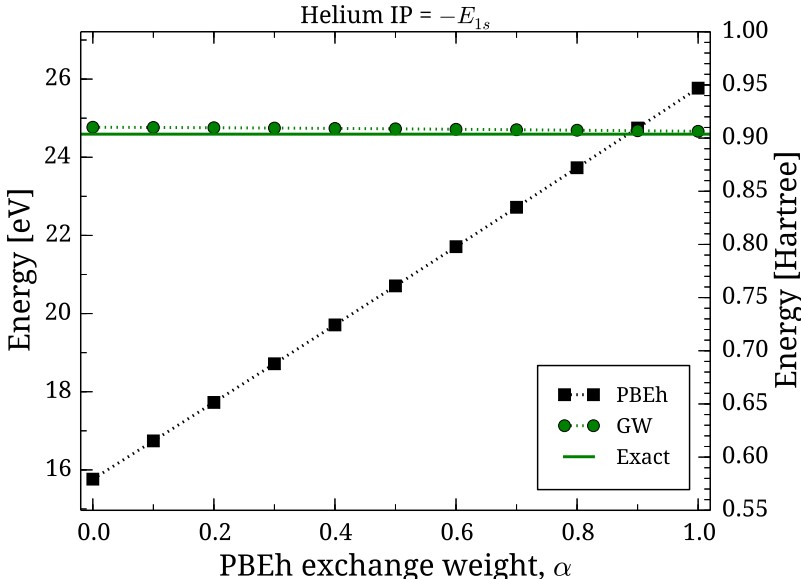

Figure 7: Starting point dependence, with respect to the PBEh exchange weight $\alpha$, of the ionization potential IP $= -E_{1s}$ as calculated in PBEh (black squares) and in the *GW* approximation (green circles), compared to the exact [32] value (green solid line).

for a comparison with TDHF and quantum chemistry methods like CI. A dependence on the starting point for *GW* and BSE calculations should be expected, although in this work we have performed a partial self-consistent *GW* concerning only the energies. In this section we will analyze the dependence of both *GW* and BSE results with respect to the starting point. We have chosen the hybrid DFT/HF PBEh approach [27] with variable exchange weight $\alpha$ [79] because this will allow to explore a full range of situations. From pure DFT-PBE at $\alpha = 0$, to a HF approach including correlation in the form of the local potential associated to the PBE DFT functional at $\alpha = 1$. The results are reported in Fig. 6(a) for the 2*S* excitation energies (both singlet and triplet), Fig. 6(b) for the 2*P*, and 6(c) for the 3*S*. It is quite surprising to see that in all the cases the value of $\alpha$, that is the starting point, is little affecting the *GW* HOMO-LUMO+n gaps, although PBEh gaps are strongly affected. This is also what we observe if we consider the ionization potential (IP), equal to minus the energy of the HOMO 1*s* state (Fig. 7). However we point out again that we performed a self-consistency on the *GW* energies. Wave functions on the other hand are kept at the level of PBEh, and these can have a more important effect on the matrix elements of the BSE kernel, and consequently also on the BSE eigenvalues. We observe such an effect in Figs. 6(a-c), in particular more on the triplet states, while singlet states seem to follow the trend of *GW* gaps. A much more important effect is to be expected on oscillator strengths since the latter are only sensitive to wave functions. This is indeed what we observe in Fig. 6(d) for the oscillator strength of the transition $1^1S \rightarrow 2^1P$, varying in a broad range, from $f = 0.51$ at $\alpha = 0$, to $f = 0.27$ at $\alpha = 1$

In conclusion, if the value of $\alpha$ and the starting point seems to affect little the result of the *GW* gaps, and in part also the energy of singlet excitations, a choice of an $\alpha$ close to 1 seems to provide results more in agreement with the exact calculation. This in particular for the oscillator strength but also for the energy of triplet states, and finally also for the ionization potential. This seems to indicate that HF is the best starting point for many-body perturbation theory calculations, at least in the case of the helium atom and probably also of other isolated

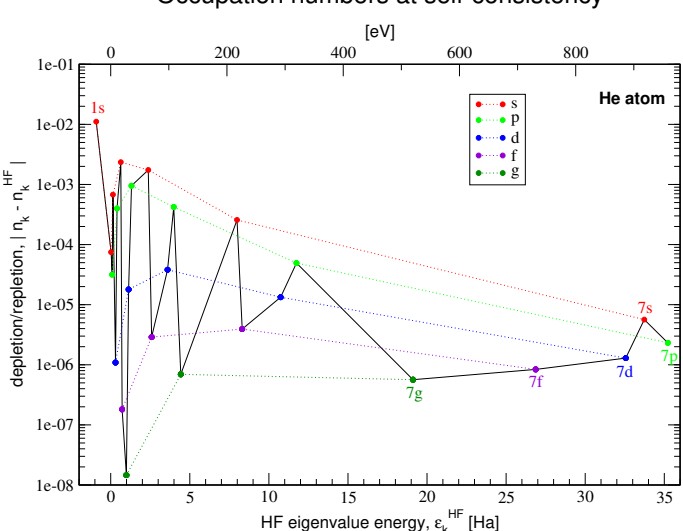

Figure 8: Depletion and repletion of occupation numbers as calculated in r-RPA (renormalized RPA) towards SCRPA taken at self-consistency, as a function of the Hartree-Fock energies. The orbital character of the states is indicated by different colors and as label of dots. The zero of energy is set to the first ionization onset.

systems.

We also report on a $GW$+BSE calculation starting from DFT LDA: the results are close to the ones starting from DFT PBE (PBEh functional at $\alpha = 0$). When starting from DFT LDA, excitation energies are 0.2∼0.3 eV larger than when starting from DFT PBE, like also the oscillator strength $f_{1^1S \to 2^1P}$, larger by 0.03 (see Fig. 6(d)).

Finally, we would like to compare our results with the available literature. To the best of our knowledge, on helium atom there are no BSE calculations, only $GW$ calculations of the ionization potential, IP $= -E_{1s}$, and these are one iteration $G_0W_0$ calculations starting from DFT LDA [80] or PBE [81]. Without performing any self-consistency, our fully dynamical contour-deformation $G_0W_0$ calculation at the $d$-$aug$-cc-pV5Z Gaussian basis set provide an ionization potential of 23.57 eV when starting from LDA, and 23.40 eV when starting from PBE. When starting from PBE, the best result Van Setten *et al.* [81] have obtained is 23.48 eV, either using the codes FHI-AIMS (their analytic continuation 16 parameters Padé approximant, P16 result) or TURBOMOLE (their no-resolution of identity, noRI result) in both cases using a def2-QZVP Gaussian basis set, which is less converged with respect to our $d$-$aug$-cc-pV5Z. By using the same def2-QZVP basis set we were able to reproduce their same result: 23.4769 eV. Van Setten *et al.* also quote a plane waves $G_0W_0$ result by the BERKELEY-$GW$ code using a plasmon-pole model and again starting from PBE: 24.10 eV, that is 0.6 eV larger than the Gaussian basis result. This result is very close to the Morris *et al.* [80] result of 24.20 eV obtained by a $G_0W_0$ on top of DFT LDA, using plane waves and with a full treatment of the frequency dependence, i.e., without using the plasmon-pole model. We remark that our $G_0W_0$ result starting from LDA is also larger (by 0.17 eV) than the $G_0W_0$ starting from PBE. So, our data seem coherent with the data available in the literature, in the limit of the expected differences between using localized and delocalized basis sets.

## 4.5 Renormalized RPA (r-RPA) and single quasiparticle energies

We will now present the self-consistent results of our r-RPA calculation. Three or four iterations were necessary to achieve self-consistency at the accuracy we quote in our tables. In Fig. 8 we

Table 9: He electron removal (first line) and addition (following lines) energies (Ha) in HF, *GW*, exact [32] and experimental (EXP) result and renormalized RPA (r-RPA). The zero of energy is set to the first ionization onset, so that the ground-state value is coincident with minus the ionization potential.

| $nl$ | HF | *GW* | Exact | r-RPA |
|---|---|---|---|---|
| $1s$ (= −IP) | −0.9179 | −0.9075 | −0.9037 | −0.9123 |
| $2s$ (= −EA) | +0.0217 | +0.0213 | > 0 | +0.0202 |
| $2p$ | +0.0956 | +0.0944 | | +0.0935 |
| $3s$ | +0.1394 | +0.1369 | | +0.1370 |

show the values of depletion/repletion of the correlated r-RPA occupation numbers, $|n_k - n_k^{\mathrm{HF}}|$, with respect to the integer uncorrelated HF occupation numbers, as a function of the HF single-particle energy. Our data were calculated by Eq. (12) and Eq. (13) and include both $S = 0$ spin singlet and $S = 1$ triplet contributions. We remark that the correlation corrections to the occupation numbers are small, 1% for the only occupied $1s$ level, becoming smaller and smaller for the unoccupied ones with increasing principal quantum number $n$. In the plot we are also able to reveal a decreasing trend at increasing angular momentum $l$. In the jellium metallic spheres studied by Catara *et al.* [16] depletions and repletions were found much larger, beyond 30% in some cases. As already indicated in the literature [82–84], the absolute value of depletions and repletions in occupation numbers and momentum distributions, can be considered a reliable indication of the correlation strength in one system.

In Table 9 we report the calculated r-RPA single-particle energies, as calculated by solving the single-particle Schrödinger equation (9), using the mean-field potential Eq. (10) calculated with the fractional correlated occupation numbers already plotted in Fig. 8. We report the values at self-consistency and compare them to the values calculated with other approaches as HF, *GW*, and the exact values only where known, in practice just only the ionization potential can be derived from an exact Hylleraas calculation. Focussing on the last occupied $1s$ energy, we see that the 1.6% error of HF is reduced to less than 1% in r-RPA, showing the same correct trend as the *GW* correction which reduces the error to 0.4%. The Hylleraas calculation cannot provide the exact values of the electron affinity and other addition energies, but comparing the r-RPA values to HF and *GW* we see that, with respect to HF, they go in the same direction of *GW* corrections, and go even beyond them. They are anyway very close to *GW* quasiparticle energies. So, the correlation corrections brought by both r-RPA and *GW* on top of the HF electronic structure seem to go in the same direction, although it is, *a priori*, not clear how they are physically related to each other. We may clarify this point in a future publication. We remark in particular that all the HOMO-LUMO+n gaps close down from HF, and r-RPA situates half way with respect to *GW*.

In Fig. 9 and Table 10 we report on the excitation energies obtained at self-consistency by the r-RPA approximation. We distinguish the case of updating only the occupation numbers [Eqs. (12) and (13)] keeping the energies at the level of HF (indicated in the table and in the figure as "r-RPA occ. only"), from the full r-RPA, where we update occupation numbers and energies [Eq. (9), indicated in figures and tables as "r-RPA occ. & ene."]. In all the cases we report the result at self-consistency. By looking at Fig. 9 and Table 10 we see that the introduction of correlated occupation numbers (r-RPA occ. only) systematically increases the energy of all excitations with respect to standard RPA. This results in an overall worsening of the excitation spectrum with respect to the exact. Renormalized RPA (r-RPA) improves only on the first $2^3S$ excitation whose energy is the only one underestimated by standard RPA. This result is rather discouraging since, when looking at the full SCRPA matrix $\mathcal{S}$, it can be expected that the introduction of fractional occupation numbers should play a major role in SCRPA.

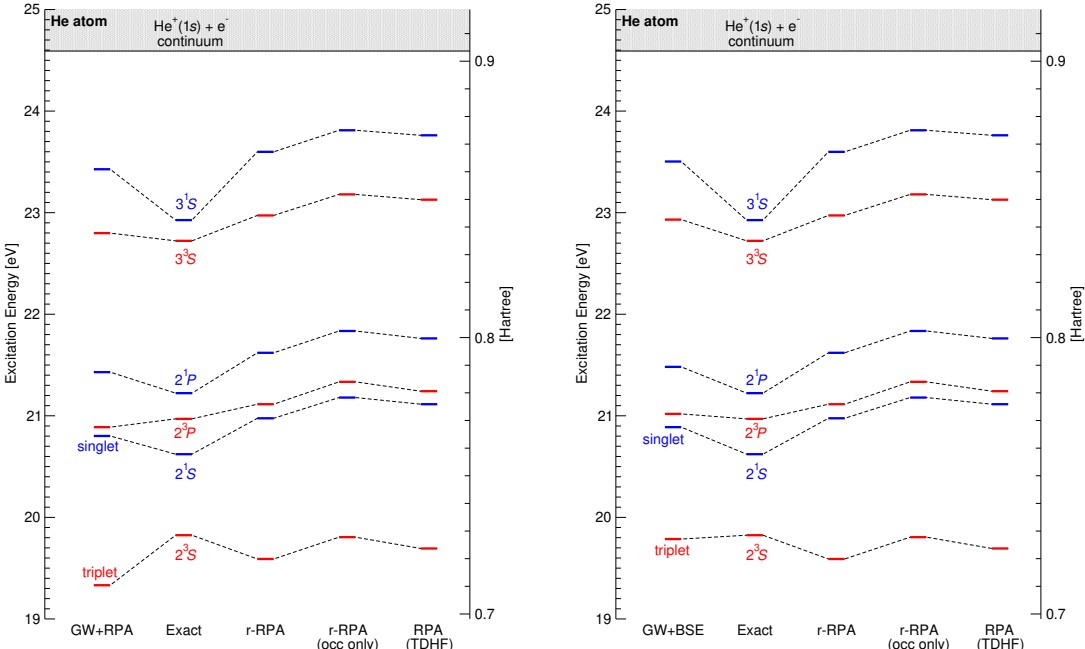

Figure 9: He atom excitation spectrum by the renormalized r-RPA towards SCRPA, in two different flavors: updating up to self-consistency the occupation numbers only (r-RPA occ. only); and updating both occupation numbers and single-particle energies (r-RPA *tout court*). We compare the r-RPA spectra to the standard RPA, to the exact solution and finally also to a *GW*+RPA unscreened kernel approximation (left panel) and to a *GW*+BSE screened kernel calculation (right). The zero of energy is set to the ground state $1^1S$.

However the situation is completely reversed when considering a full r-RPA, taking into account corrections to the occupation numbers and also to energies (r-RPA occ. & ene. or r-RPA *tout court*). The effect of replacing occupation numbers in energies, that might appear second-order with respect to their direct effect when replaced where they appear in the $\mathcal{S}$ matrix, is instead quite important to correct standard RPA towards the right direction. We see that the excitation energy is reduced not only with respect to the r-RPA occ. only approximation, but also with respect to standard RPA. This results in an overall improvement with respect to standard RPA, towards the exact solution. Again the exception is represented by the first excited state where in full r-RPA we observe a worsening.

These r-RPA results can be better understood if compared not directly with the *GW*+BSE approach, but rather with a *GW*+RPA calculation using a $\bar{\nu}$ unscreened kernel. Indeed, in both the full r-RPA and the *GW*+RPA cases the novelty with respect to standard RPA (TDHF) is the introduction of a correlated, in place of the uncorrelated HF electronic structure, as starting point of the RPA equations. While the kernel keeps in all cases the same $\bar{\nu}$ as in standard RPA. We see in Fig. 9 (left panel) and Table 10 that, with respect to standard RPA, the effect of both *GW*+RPA and r-RPA is exactly in the same direction. For all excitations we observe a reduction of their energy with respect to standard RPA. This can be directly traced back to the reduction of single-particle HOMO-LUMO gaps taken as starting points to the same $\bar{\nu}$ kernel RPA equations. The *GW*+RPA excitation energies are lower than r-RPA simply because the HOMO-LUMO *GW* gaps are smaller than r-RPA. For this reason the *GW*+RPA is more in agreement with the exact result, again with the exception of the first $2^3S$ excitation where both r-RPA and *GW*+RPA go in the wrong direction with respect to standard RPA, and the more important *GW*+RPA correction turns out in a worse result.

Table 10: He excitation energies in atomic units (Hartree, top) and electronvolt (eV, bottom) as calculated in a renormalized RPA towards SCRPA, both updating only the occupation numbers, or also the energies. The zero of energy is set to the He ground state $1^1S$.

| $n^S L$ | RPA (TDHF) | r-RPA occ. only | r-RPA occ. & ene. | **Exact** | $GW$ + RPA | GW + BSE |
|---|---|---|---|---|---|---|
| | | | atomic units [Ha] | | | |
| $2^3S$ | 0.7237 | 0.7278 | 0.7199 | **0.7285** | 0.7104 | 0.7271 |
| $2^1S$ | 0.7759 | 0.7783 | 0.7708 | **0.7578** | 0.7644 | 0.7676 |
| $2^3P$ | 0.7806 | 0.7840 | 0.7759 | **0.7706** | 0.7676 | 0.7724 |
| $2^1P$ | 0.7997 | 0.8024 | 0.7945 | **0.7799** | 0.7875 | 0.7894 |
| $3^3S$ | 0.8499 | 0.8518 | 0.8442 | **0.8350** | 0.8378 | 0.8427 |
| $3^1S$ | 0.8732 | 0.8750 | 0.8672 | **0.8425** | 0.8609 | 0.8637 |
| | | | electronvolt [eV] | | | |
| $2^3S$ | 19.692 | 19.805 | 19.590 | **19.824** | 19.330 | 19.786 |
| $2^1S$ | 21.115 | 21.178 | 20.976 | **20.621** | 20.800 | 20.888 |
| $2^3P$ | 21.242 | 21.333 | 21.112 | **20.969** | 20.888 | 21.018 |
| $2^1P$ | 21.762 | 21.835 | 21.619 | **21.222** | 21.428 | 21.480 |
| $3^3S$ | 23.128 | 23.179 | 22.973 | **22.722** | 22.798 | 22.930 |
| $3^1S$ | 23.762 | 23.811 | 23.598 | **22.926** | 23.426 | 23.502 |

To correctly describe this first excitation and improve, rather than worsen with respect to standard RPA, a correction of the kernel seems required. For example the introduction of screening into the bare particle-hole exchange interaction of the RPA kernel, like done in BSE. This reduces the strength of the kernel and so of the correction to the excitation energy when starting from $GW$+dRPA. The effect of the screened BSE $W$ kernel is impressively evident on this first $2^3S$ excitation (compare left and right panel of Fig. 9). The screening reduces its effect when moving to higher excitations. For the highest excitations one might argue that the introduction of the screening, although with smaller and smaller effect, goes in the wrong direction to increase the energy, but we remind that the overestimation of the excitation energy is a finite basis set effect due to the poor representation of highly delocalized states by Gaussians also detected in the CI calculation.

The r-RPA result here presented may appear not yet satisfactory, for example if compared to BSE. However, we think it is a very encouraging result. This result makes us hope that the introduction of the two-particle correlation terms into the full SCRPA $\mathcal{S}$ matrix can reduce the strength of the kernel, like it happens in BSE when introducing the screening into the bare Coulomb $v$. Indeed, the neglect of the correlation terms in $\mathcal{S}$ atrophies SCRPA very much. This the more so as the correlation terms can be shown to contain screening in a similar way as with BSE. These aspects may be elaborated in a future publication.

In Table 11 we report the $f_{1^1S \to 2^1P}$ first dipole allowed transition oscillator strength for r-RPA. We remark an improvement with respect to standard RPA. This is mostly due to the update of occupation numbers. Since the oscillator strength is above all sensitive to wave functions, the difference between r-RPA with or without updating the energies is less evident than in excitation energies themselves. Nevertheless, the fact to have different energies along the diagonal of the RPA $\mathcal{S}$ matrix has also an effect on eigenvectors, wave functions and, thus, oscillator strengths. This effect is also appreciable when comparing the standard RPA to the $GW$+RPA oscillator strength. The latter even shows a worsening. A correction to occupation numbers and/or the kernel, like in BSE, is required to improve the oscillator strength towards

the good direction.

Table 11: Helium atom first dipole-allowed $1^1S \to 2^1P$ transition oscillator strength, calculated in RPA (TDHF), r-RPA updating up to self-consistence occupation numbers only, r-RPA updating both occupation numbers and energies, exact Hylleraas calculation [32], RPA on top of $GW$ quasiparticle energies, and BSE.

|  | RPA (TDHF) | r-RPA occ. only | r-RPA occ. & ene. | **Exact** | $GW$ + RPA | GW + BSE |
|---|---|---|---|---|---|---|
| $f_{1^1S \to 2^1P}$ | 0.2916 | 0.2889 | 0.2877 | **0.27616** | 0.2946 | 0.2763 |

# 5  Conclusions

Our work presented a comparison on the same footing, in particular using the same Gaussian basis set, of several many-body approaches, including a not so much explored renormalized RPA (r-RPA) derived from the EOM method developed in nuclear physics. Our work shows that the r-RPA, which is a sub-product of the SCRPA approach, improves over the standard RPA (i.e. linearized time-dependent Hartree-Fock (TDHF) [33]) and achieves a result of accuracy comparable to $GW$+BSE, except for the first excited state where there is no improvement. Also $GW$+BSE improves on dRPA on top of both HF and $GW$, but also on RPA/TDHF. CI is certainly one of the most accurate methods, but localized-basis-set issues seriously reduce its accuracy on the highest excited states, well outside chemical accuracy. On the Rydberg series, an Exact-DFT+TDLDA calculation done in real space shows superior performances with respect to even Gaussian-basis CI. In the same CI Gaussian basis set, we have presented also the DFT-LDA+dRPA and DFT-LDA+TDLDA helium excitation spectra, arguing that the question of the boundness of the Rydberg series depends on the way the ionization potential is calculated. On the ground state CI achieves chemical accuracy, but cannot do better even relying on recent basis set extrapolation techniques. On the other hand, standard QMC, Slater-Jastrow variational Monte Carlo (VMC) followed by diffusion Monte Carlo (DMC) at the actual computer power, has shown 2 orders of magnitude superior accuracy with respect to CI. We should mention however that the helium ground state wave function is nodeless, a favorable case where QMC is unaffected by the so-called fermion sign problem.

# 6  Acknowledgments

We thank Xavier Blase, Ivan Duchemin, and Markus Holzmann for useful discussions.

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
