# Peer review of "Comparing many-body approaches against the helium atom exact solution"

_SciPost Physics, doi:SciPost Phys. 6, 040 (2019)_

## Round 2 · Referee Report · Anonymous · 2019-2-13

Strengths

1-clear and pedagogical
2-insightful
3-of broad interest

Weaknesses

1-some of the conclusions valid for the specific case of Helium atom might not necessarily apply to larger systems

Report

In this work the authors use the simple but realistic case of He atom to compare the quality of various methods and approximations used in condensed matter physics, nuclear physics, and quantum chemistry. The He atom has two electrons, hence electron correlation (which is in general hard to describe in all commonly used theories) is well displayed; moreover the system can be solved exactly, and hence be used as a benchmark.
The manuscript is well written, very pedagogical, and self-contained. I enjoyed very much reading it, since various abstract concepts, such as the correction to the Kohn-Sham kinetic energy contained in the exchange-correlation energy functional of DFT, becomes tangible by showing the numbers for the case of He.

I have only a couple of questions/comments:

1) If I understand well Eq. (3) is an ansatz. Maybe the authors should explicitly mention this.
2) Eq. (11) calculates both the resonant and antiresonant excitation energies. Does the sum over the excitation energies in the correlation energies (first equation of page 7, first column) run over both resonant and antiresonant energies? Maybe it would be useful to write it in the text.
3) The second equation of page 7 (first column) seems to imply that the correlation energy comes from the coupling between resonant and antiresonant energies. Is that correct? Is there a simple way to see this?
4) I found the renormalized RPA and SCRPA equations very interesting. The difference with the usual RPA/TDDFT/BSE equations is the presence of fractional occupation numbers. How do the authors define what is a particle or a hole, and hence the particle-hole basis set for the matrix equation (11) if occupation numbers are fractional? Do they fix them to the HF initial values of the occupation numbers?
5) The limits of some methods, such as CI, because of an incomplete basis set might be mitigated for larger systems. In medium-sized or large molecules a moderately large basis sets can be quite adequate because of the effect of basis set sharing, i.e. the fact that each atom profits from the basis functions on its many neighbors. Moreover some methods or approximations can be less dependent on basis set size (for example, in general wavefunction-based methods show a slower convergence with respect to the basis set size than DFT; or hybrid functional are more sensitive to the basis set size than pure functionals, see, e.g J. Chem. Phys. 121, 7632 (2004)). Finally the aug-cc-pVXZ family of basis sets have been developed for correlated wave function methods and are not optimum for other methods such as DFT (see, e.g., J.Phys.Chem.A 121, 6104 (2017); this hence plays a role into the comparison between CI and DFT/Green’s function-based methods results. It would be nice if the authors could comment on this in the paper.
6) In the second column of page 12 the authors mention “the exact DFT kinetic energy”, which has nothing to do with the exact kinetic energy. What do they mean for exact DFT kinetic energy is the exact KS kinetic energy, as they point out a few lines later. I think this can be confusing and it would be better to call it exact KS kinetic energy from the start. In principle the exact DFT kinetic energy as functional of the density is the exact kinetic energy.
7) The TDLDA calculations on top of an exact DFT KS spectrum seems similar in spirit (although not in the physics) to the rigid shift (scissor operator) that one often uses in solids. Is that correct? Maybe the authors could mention this in the manuscript.

In conclusions, I find this manuscript very interesting, the physics valid, and targeting a broad audience. I recommend publication as a regular article in SCIpost after minor corrections according to the comments above.

Requested changes

See my comments 1-7 of the report

  • validity: high
  • significance: high
  • originality: high
  • clarity: top
  • formatting: excellent
  • grammar: excellent

Author:  Valerio Olevano  on 2019-03-01  [id 453]

(in reply to Report 1 on 2019-02-13)

We thank Referee 1 for the very high marks and the positive appreciation of our work and paper. Below we answer all the points raised. 1) Actually Eq. (3) is more than an Ansatz. In its generally accepteded definition, an Ansatz is a guess, following an assumption or conjecture, to provide an approximate result as close as possible to the exact solution without the requirement to have all the degrees of freedom to cover the exact solution. The series we wrote there, including also negative powers, mathematically represents a "formal solution" to the He Schroedinger equation. This is discussed in our answer to Referee 2 where she/he questions the exactness of Hylleraas-like solutions (points 3 and 4). We invite Referee 1 to comment on Referee 2's and ours arguments there. 2) The sum runs only over resonant (positive) energies, $\Omega_\lambda > 0$. This is the case also for the following equation (the second equation on page 7, first column), where it is more evident since $\Omega_\lambda$ appears accompanied by the only-resonant Tamm-Dancoff approximation (TDA) $\Omega_\lambda^\mathrm{TDA}$. We changed the text and indicated this fact by restricting the sum to run over $\lambda > 0$, where $\lambda < 0$ denotes the eigenvalue indices of negative energies. 3) Correct. The TDA builds correlations only into the excited states, whereas the ground state remains uncorrelated. One must go beyond the TDA to build correlations also into the ground state. The simplest way to see this is to observe that the TDA approximation comes out if we approximate the $\hat{Q}^\dagger_\lambda$ operators not only to be of the one-body form as in Eq. (7b), but also by setting $k_1 = p$ particle (conduction) state and $k_2 = h$ hole (valence) state. Indeed, in the secular equation [Eq. (8)] we then have only the $A$ component for the matrix $\mathcal{S}{ph,p'h'}$, without the coupling $B$ component, which is the TDA approximation. Now the ground state corresponding to this TDA approximation, that is the ground state killed by these approximate $\hat{Q}\lambda = \sum_{ph} \chi^{\lambda *}{ph} \hat{c}^\dagger_h \hat{c}_p$ operators (i.e., $\hat{Q}\lambda | \Phi_0 \rangle = 0$), is nothing else than the uncorrelated Hartree-Fock (HF) ground state $| \Phi_0^\mathrm{HF} \rangle$. By removing this constraint one can go to random-phase approximation (RPA) equations beyond the TDA and have a correlated ground state beyond Hartree-Fock. This is explained in more depth in Ring and Schuck, Sec. 8.4 and sections around. 4) If $N$ is the number of electrons in the system and $\epsilon_k$, $\phi_k(r)$, and $n_k$ are, respectively, the HF or quasiparticle energies, wave functions, and occupation numbers, we denote as hole $h$ (valence) states those for which $k\le N$, and as particle $p$ (conduction) states those for which $k>N$. In our spin-unpolarised calculation for He, $N=1$, so that there is only one hole state. We start from the HF uncorrelated state ($n_h = 1$ and $n_p =0$), but following the self-consistent calculation they are depleted, $n_h\le 1$, or repleted, $n_p \ge 0$. If we were solving the equation in the Bethe-Salpeter equation (BSE) form, $L_{k_1,k_2;k_3,k_4}(\omega) = L^0_{k_1,k_2;k_3,k_4}(\omega) + L^0(\omega) \Xi L(\omega)$, at any self-consistent step we would reinject the fractional occupation numbers $n_k$ into $L^0$:

\[ L^0_{k_1,k_2;k_3,k_4}(\omega) = (n_{k_1} - n_{k_2}) \frac{\delta_{k_1 k_3} \delta_{k_2 k_4}}{\omega - (\epsilon_{k_2} - \epsilon_{k_1}) \pm i \eta}. \]

In the RPA form of Eq. (11), or more compactly $H^\mathrm{exc} | \Psi_\lambda \rangle = \Omega_\lambda | \Psi_\lambda \rangle$, the previous replacement is reflected in the matrix $H^\mathrm{exc}$:

\[ H^\mathrm{exc}_{k_1,k_2;k_3,k_4} = (\epsilon_{k_2} - \epsilon_{k_1}) \delta_{k_1 k_3} \delta_{k_2 k_4} - i \sqrt{n_{k_2} - n_{k_1}} \Xi_{k_1,k_2;k_3,k_4} \sqrt{n_{k_4} - n_{k_3}}, \]

where now in rRPA there explicitly appear the occupation number factors under square roots which are absent (being equal either to 1 or to 0 depending on whether the $k_i$ are $p$ or $h$) in the standard RPA or BSE case. So, in rRPA or in SCRPA they must not be fixed to integer uncorrelated HF values. The $H^\mathrm{exc}$ can then be developed in its $A$ and $B$ components. 5) Yes, we also think there should be a mitigation in large molecules, at least for the ground and the lowest excited states, while we still see difficulty reproducing the Rydberg series. However, this remains only an unproven hypothesis in the absence of an exact solution to compare with. One should expect that, on localised Gaussians, "long-range" $1/r$ methods (e.g., wave function based, GW, hybrids) converge more slowly than short-range $e^{-r}$ methods [e.g., local density approximation (LDA) exchange-correlation potential or other pure density functional theory (DFT) functionals]: this is the reason why we chose basis-set families optimised for correlated wave-function methods. However, if one checks Fig. 1 of J. Phys. Chem. A \textbf{121}, 6104 (2017), one can see that for He the convergence error can be reduced from $10^{-4}$ to $10^{-6}$ Ha using basis-set families optimised for DFT-LDA. Since the error of the LDA with respect to the Hylleraas result is already $7 \cdot 10^{-2}$ Ha, the basis-set convergence error of $10^{-4}$ can be neglected. We added all these interesting comments to the paper. We thank the referee for raising this point. 6) We fully agree. We changed "exact DFT kinetic energy" to "exact KS kinetic energy". We really appreciate this remark since we wish to be as rigorous as possible. 7) Yes, if one compares the time-dependent local density approximation (TDLDA) on top of exact-DFT with the TDLDA on top of DFT-LDA (2nd and 4th columns in Fig. 5) there is an evident $\sim 4$ eV shift, but it does not look really rigid. However it is true that the DFT-LDA+TDLDA excitation spectrum could be improved by applying a rigid scissor operator of $\sim 4$ eV to DFT-LDA Kohn-Sham (KS) eigenvalues, such that the KS HOMO-LUMO gap could get closer not to the real HOMO-LUMO (electron affinity minus ionisation potential) gap of the system, but to the optical gap, e.g., the 2S excitation energy in He and the optical onset of a solid. We added this discussion, which further enriches our manuscript. We thank the referee for this remark.

---

## Round 2 · Referee Report · Anonymous · 2019-2-18

Strengths

- clear, pedagogical and self-contained
- of general interest
- scRPA and rPRA parts are genuinely new

Weaknesses

- a bit verbose
- discussion in the introduction about the "exact" wave function of the He atom is very misleading.
- sometimes the claims are too strong

Report

In the present manuscript, Li et al. report an exhaustive study of the helium atom using electronic structure methods ranging from RPA-type methods to more conventional methods such as QMC, CI and TD-DFT.
I am to admit that, at first sight, I was a bit scared by the length of the paper, but I have learnt to like throughout this pleasant reading.
In my humble opinion, I still believe that the present manuscript is too long but every single author has his/her own style, and you have to accept this I guess.
Although the He atom has been studied to death in the last century, I believe that the present manuscript is still very interesting as it studies new, unconventional methods.
I have, however, a long list of comments that the authors should consider.

- abstract: the helium atom is indeed a real system but, in chemistry at least, this is what ressemble the most a model.

- abstract: studying methods on equal footing is not a trivial task.
The authors have chosen to use the same basis set throughout their study.
This is one way of doing it but I don't think this is the best.
My reason for saying this is that each method has its own sensitivity to basis set incompleteness. For example, density-based methods are known to converge faster wrt the size of the one-electron basis set than CI or CC methods. Therefore, a 6-31+G* B3LYP calculation might be much more converged (in a basis set sense) than a CCSD/aug-cc-pVTZ.
My belief is that one should always work in a near complete basis to assess faithful quantum chemistry methods.
The authors should at least comment on this.
Moreover,

- abstract and Introduction: sadly, we don't have the exact solution for the He atom.
The best we have is a very accurate numerical solution.
The wave function of the He is non-analytic and cannot be obtained in closed form.

- I strongly disagree with the authors on the "exact" solution of the He atom.
There is no mathematical guarantee that an expansion in terms of Hylleraas coordinates does converge.
As shown by many authors, one must introduce (non-analytic) logarithmic terms to ensure that the series converge to the right limit.
An expansion in terms of Hylleraas coordinates yields an analytic function.
As correctly mentioned by the authors, the exact wave function of He is non-analytic, therefore one cannot get the exact He wave function with Hylleraas coordinates.
Nakatsuji has shown that one must insert non-analytic terms to ensure a proper convergence.
Actually, Bartlett has shown that one cannot find a solution using Hylleraas variables within the Frobenius method as one hits a contradiction very quickly (https://journals.aps.org/pra/pdf/10.1103/PhysRevA.30.1506).
One could also use the Fock expansion but, as shown by Morgan and coworkers, the radius of convergence of such an expansion is zero (https://link.springer.com/content/pdf/10.1007/BF00526420.pdf).
Therefore, there is no guarantee that the numerical calculation converges to the right value.

- The authors mentioned that the present study is much better than conventional models but they do not cite any.
Also, as mentioned above, the helium atom is indeed a real system but, in chemistry at least, this is what ressemble the most a model.
It would be interesting to mention the pros and cons of each model (helium, Hubbard, spherium, etc) as, for example, in J. Chem. Theory Comput. 14, 3071 (2018).

- Page 2: "In short, it consists of self-consistently re-injecting into the parameters of the RPA equations the correlations from the eigensolutions out of the RPA equations themselves. An obvious place where in the standard RPA equations correlations are disregarded is in single-particle occupation numbers."
It would be nice to make that statement clearer. Thankfully, this is the short version...

- Page 3: "Although the DFT- LDA + TDLDA helium-atom excitation spectrum has been discussed several times in the literature, numerical results have never been published to our knowledge."
Additional references would be welcome.

- Page 4: Expansion (3) is only suitable for S states (singlet and triplet).
However, it is not suitable for higher angular momentum (P, D, etc) where one needs to use a different ansatz.

- Page 5: Is it really appropriate to report Eq. (4) as this type of expansions is not used in the present manuscript?
Moreover, the addition of logarithmic terms in the wave function has been already mentioned earlier in the manuscript.

- Page 5: The second sentence of Sec. III needs to be modified.

- Page 5 and 6: for each formula involving summation of RPA quantities, it would be very useful to know if the summation runs over singlet and/or triplet excitations (and why).

- Last equation in left row of page 5: typo with two equal signs.

- Equations (12) and (13): very interesting part. I always wanted to know more about rRPA and scRPA, and it is very well explained here. However, it would be nice to have the exact definition of the quantity \chi.
It must be related to the eigenvectors (X+Y) of the RPA problem but it's better to be sure that's indeed the case.

- Page 8 Sec. C: These days, selected CI methods (like CIPSI) can be used to get near-FCI energies for ground and excited states of molecules with several heavy atoms.
See, for example, J. Chem. Phys. 147, 034101 (2017) or J. Chem. Theory Comput. 14, 4360 (2018) or 10.1021/acs.jctc.8b01205
One can even do DMC on top of it in order to complete the basis set
See, for example, J. Chem. Phys. 149, 034108 (2018).
I believe that it should be mentioned as the present claim isn't fair.

- Page 8: I am very surprised that the authors cannot perform a proper FCI calculation on He with d-aug-cc-pV5Z.
It's only a CISD with 2 electrons and 115 orbitals. I'm pretty sure it can be performed with GAMESS in a few minutes.

- Page 8 Sec. D: The problem with RPA/GW methods is that the combination of things that can be done is just infinite, and one gets lost very quickly. Could the author provide a clear diagram summarising which methods they use to get which quantity?
GW is already using dRPA to get the neutral excitations used to construct the self-energy.

- Page 18: The depletion/repletion of the occupation number is small because it is a single-reference system (the HF wave function is a very good approximation).
In metallic systems, as mentioned by the authors, the situation is drastically different as they are usually strongly mutli-reference if one uses a localized basis set.

Requested changes

See my points above.

  • validity: high
  • significance: high
  • originality: high
  • clarity: high
  • formatting: good
  • grammar: reasonable

Author:  Valerio Olevano  on 2019-03-01  [id 454]

(in reply to Report 2 on 2019-02-18)

We thank Referee 2 for the positive appreciations. We realised too late that the paper went outside reasonable length limits. Below we comment on all the points raised by the referee.

1) (abstract) and 5) (Model vs Real) We are surprised to learn that in chemistry the helium atom is considered a model. In physics, third-year students, after having studied quantum mechanics (QM) and applied it to the quantum barrier, the square-well, and the harmonic oscillator models, then study the important verification of QM on real systems, that is against experiment and reality: first the hydrogen and then the helium atom. Also historically, after Bohr had demonstrated that QM was working for H, he called for an effort to find at least another verification of QM on a system with more electrons. Hylleraas answered and his solution for He in good agreement with experiment constituted an important verification of QM as a "Universal" theory, i.e. not only working on only one system (in a period where QM was under the attacks of Einstein, Schroedinger and his cats, etc.). Physics students are educated by atomic physics university textbooks, like the popular book by Bransden and Joachain, about the importance of this QM theory-experiment verification by Hylleraas (see Chap. 6 and in particular Tables 6.5 and 6.3). So, in the minds of all us physicists, helium is a very real system and an important verification of the QM theory against experiment and reality.

2) (abstract) Yes, we fully agree. The same issue was already raised by Referee 1, point 5); please see our answer there. We commented about this important methodological question in the paper.

3) (abstract and Introduction) and 4) It has been mathematically proved [Kato, Am. Math. Soc. {70}, 212 (1951)] that solutions to the He Schroedinger differential equation, under the rigorous boundary conditions in Eq. (11) of [Kato, Am. Math. Soc. {70}, 195 (1951)], exist. Let's call these exact solutions $E^\mathrm{He}i, \Psi^\mathrm{He}_i$. The ground-state energy $E^\mathrm{He}_0$ is then a well-defined real number. Kato's theorems also state that $\Psi^\mathrm{He}_i$ are continuous everywhere, even at the singularities of the Coulomb potential $r_1=r_2=r = 0$. An "exact numerical solution" can be defined in a mathematically rigorous way, but we prefer to state here a more intuitive definition: it is a numerical method that provides a well-defined numerical estimate $E$ of the exact solution and a well-defined error bar $\Delta E$ (or, equivalently, an indication of the significative digits), which therefore provides both an upper and a lower bound, and so an interval within which the exact solution $E^\mathrm{He}$ certainly falls. And it is possible to systematically improve the estimate by reducing the error bar or increasing the number of significative digits. These are exactly the characteristics owned by the Hylleraas(-like) approaches, as can be appreciated by looking at the history of this solution as presented in Table I of Nakashima and Nakatsuji [J. Chem. Phys. {127}, 224104 (2007) http://doi.org/10.1063/1.2801981 ], or in Fig. 1 of Schwartz [Int. J. Mod. Phys. E {15}, 877 (2006), http://doi.org/10.1142/S0218301306004648 ]. On the other hand, we must recognise that none of the methods we presented in our paper owns these characteristics, apart from diffusion Monte Carlo in the nodeless case of the ground state of He. It is impossible to estimate the error associated, e.g., to the $GW$ approximation, i.e. the contributions due to missing diagrams, configurations, etc., and it is impossible to estimate the error due to the finite size of the basis-set. Although some of these methods can be defined "Accurate", none of them is at the same level of Hylleraas, which can hence be superiorly defined "numerically exact". However we thank the Referee to let us remark an inaccuracy in our paper: we stated that the He Schroedinger equation cannot be solved "analytically" and that the Hylleraas solution is "nonanalytic". We must replace these expressions by "closed-form" since we checked that, according to mathematical conventions, a converged infinite sum (and so Hylleraas-like power series, see below) is an analytic expression though not in closed form. Finally, is the difference between "closed-form exact" and "numerically exact" solutions really so large? At the end what is closed-form and what is not reduces only to a mathematical convention. Indeed, if we had not conventionally introduced transcendental functions, the differential equation $df/dx = 1/x$ would have no solution in a closed form, so that one should be content with the numerical solution $\int_1^2 dx \, 1/x = 0.693 \pm 0.001$, instead of the compact $\int_1^2 dx \, 1/x = \mathrm{ln}(2)$. And it is very common in analysis to define new functions as the primitives or the solutions of some differential equations, for example the special functions e.g. $\mathrm{erf}(x) = const \int 0^x dt e^{-t^2}$ or the Bessel functions $J\alpha(x)$. So one can in principle as well introduce the "Helioids" $\Psi^\mathrm{He}_i, E^\mathrm{He}_i$. Also, behind the decimal representation of $\mathrm{ln}(2)$ as provided by a computer, there is always an expansion in series of powers. The problem is, in all cases, to provide a numerical decimal representation at any established digit accuracy.

4) The cited work by Bartlett et al. [Phys. Rev. {47}, 679 (1939)] shows that the original Hylleraas series cannot be a formal solution to the He Schroedinger equation, and Withers [Phys. Rev. A {30}, 1506 (1984)] extended this result also to Frobenius series (just a mathematical curiosity since so far nobody has used a Frobenius series on He). The Bartlett objection was already present to Kinoshita, the first Hylleraas follower to have significantly improved the He solution. Kinoshita discusses the objection in [Phys. Rev. {105}, 1490 (1957)], page 1491, 3rd paragraph. However he also shows that a power series in the variables $s$, $p=u/s$ and $q=t/u$ (which, unlike the original Hylleraas $t<u<s$, are really independent), which corresponds to an extension of the original Hylleraas series to negative powers of $s$ and $u$, represents a "formal solution" to the He Schroedinger equation. The demonstration is presented in App. A where he also shows that the restriction $l \ge m \ge n$, corresponding to the original Hylleraas series, cannot lead to any formal solution, thus recovering the Bartlett result. This fact is also acknowledged in the Referee-cited work of Withers [Phys. Rev. A {30}, 1506 (1984)], who states: "His (Kinoshita's) solution, like Fock's solution, satisfies the wave equation, whereas Hylleraas's series does not." The expansion Eq. (3) in our paper deliberately leaves unspecified the limits of the series, so as possibly to refer to both the original Hylleraas and also the Kinoshita series. The question of the logarithmic factor $\ln(s)$ was already present to Kinoshita, but he showed that, to have a formal solution, it is not necessarily required, though it can greatly speed up the convergence of the series. (Notice that Kato theorem states that the wave function is finite even at $s=0$, so that the logarithmic divergence must be avoided by setting to zero the coefficient $c_{jlmn} s^l ln(s)^j$ for $l=0$. At any $s>0$ the logarithm can well be expanded in power series, e.g. $\mathrm{ln}(s) = \sum_{n=1}^\infty (-1)^{n-1} (s-1)^n/n$, so that its effect can be fully taken into account by modifying the coefficients of the Kinoshita series which will then converge at the same limit. But this can slow down the convergence with respect to the number of coefficients.). As to Fock's expansion we here quote the conclusions of the cited work of Morgan [Theor. Chem. Acta {69}, 181 (1986)]: "It has been shown that Fock's expansion for S-state solutions of Schroedinger's Equation for two-electron atoms and ions has virtually optimal convergence properties. The expansion converges not only in a small neighbourhood of $R = 0$, but for all finite $R$. Better behaviour could not be expected." So, Morgan found that the radius of convergence is $>0$.

5) Thank you for this very good suggestion: spherium is a model which is very exclusively related to the helium atom and it is absolutely worth to be mentioned here. Moreover the cited paper is exactly in the spirit of our work: a comparison of many-body methods against an exact solution. Please, notice that in that paper a label "Exact" is used for quantities "obtained from near-exact calculations".

6) (Page 2) We agree that the material indicated by the referee could appear quite cryptic for material presented in the Introduction. We have removed the unclear statements, and we simply introduce SCRPA and r-RPA, like all the other methods in the Introduction, postponing their explanation to the appropriate sections.

7) (Page 3) We added the requested references which discussed, without presenting, the DFT-LDA+TDLDA results.

8) (Page 4) It is true that one first needs to introduce also angular variables, e.g. the polar angle $\theta$. But since the total angular momentum is conserved and is a good quantum number, it is straightforwardly sufficient to multiply the Hylleraas series by spherical harmonics (more precisely, Clebsch-Gordan combinations of them to form the given total $L$). For example, for P states, after having summed over $m$, it is sufficient to multiply the Hylleraas series by the first Legendre polynomial $P_l(\cos(\theta)) = \cos(\theta)$ [Schiff et al., Phys. Rev. A {4}, 885 (1971); Kono and Hattori, Phys. Rev. A {29}, 2981 (1984).]

9) (Page 5) Previously the Referee invoked the Bartlett objection to claim that the expansion of Eq. (3) is not a formal solution to the He Schroedinger equation and that, "As shown by many authors, one must introduce (non-analytic) logarithmic terms to ensure that the series converge to the right limit". Our Sec. II is meant to target the broadest interest readership of SciPost Physics, which includes not only atomic physicists, introducing them to the electronic structure of the He atom as calculated by the exact theory, which perfectly agrees with experiment. Beyond the Hylleraas and Kinoshita expansions of Eq. (3), we thought, exactly like the Referee in his or her point 4, that it is important to at least mention this logarithmic factor.

10) (Page 5) We have removed this sentence; this information is given later.

11) (Page 5 and 6) The index $\lambda$ runs over all excitations. So it is understood that it everywhere runs over both singlet and triplet excitations. We now state it clearly from the outset.

12) (Last equation in left row of page 5) The second plus sign is needed since the equation is continued in Page 6. We will correct the typographical appearance of this equation in the final editorial-required formatting of the paper.

13) (Equations (12) and (13)) Yes, the $\chi$ in Eqs. (12) and (13) is that which is defined in Eq. (8), and refers to both $X$ and $Y$. However you can see that in Eqs. (12) and (13) only the hole-particle $\chi_{hp}$ appears, so only the $Y$ part and not the $X$.

14) (Page 8 Sec. C) Our purpose here is to present only methods, such as full configuration interaction (CI), that are well known to physicists since 3rd-year atomic physics courses (e.g., CI is presented aside the Hylleraas solution in Bransden and Joachain's textbook). However if we mention ICE-CI, it is more fair to mention also CIPSI and CCSD (equivalent to FCI in He). We anyway do not enter into their description and send the reader to the cited papers. It would be very interesting if somebody in chemistry could benchmark all the methods in those papers against the Hylleraas solution.

15) (Page 8) It was not a problem of CPU time, rather of memory management which is probably better managed by GAMESS than by ORCA. We prefer not to enter into questions about which code is best. We modified "computing power" into "computing resources".

16) (Page 8 Sec. D) The requested diagram already exists: In Fig. 1 we present polarisability Feynman diagrams associated to all the approximations explored in this work. By looking at Fig. 1 many-body physicists can immediately understand what an approximation acronym refers to. Following the referee's remark we decided to add a reference to Fig. 1 in Sec. D and Page 8. We thank the referee for this suggestion. It is true that $GW$ already uses dRPA to get the neutral excitations to build $W$ and the $GW$ self-energy; this is the standard and understood way for $GW$. If we also try to introduce this information, we risk confusing the reader. We reserve dRPA and RPA only for the steps to calculate excitations.

17) (Page 18) Yes, in our full CI calculation we found that the HF determinant has a weight of more than 99%. The Referee raises a very interesting point we had not reflected on: if we used delocalised basis sets, such as plane waves, much more appropriate for metallic systems, would these systems continue to be strongly multi-reference? Are single/multi-reference characteristics physical properties of a system? Or, does the fact that it is basis-set dependent indicate that it is an unphysical property? This is an interesting question to be investigated.

---

## Round 3 · Referee Report · Anonymous (Referee 2) · 2019-3-17

Report

I am happy with the authors' answers.

---

## Round 3 · Author Response

Answers to the Editor 2nd point:
We are all researchers, and researchers are exploring the world amazed like children, and like children we must do it by playing (a fact that research financing agencies, especially European, do not understand, unfortunately). And models are our preferred toys. We are absolutely convinced about the importance of studying models: only by playing with models we can grasp the full complexity of, e.g., many-body theory. However, also like children, at a certain point we must take awareness of the limit where the game is over and the reality starts: the confusion of the two planes is not at all good (for both researchers and children). So far as a model could be a beautiful toy to be reluctantly given up, we must be aware of its limits as portrait of the reality. I find "forced" the Editor's opinion that there is no "distinction" between models and real systems.
The Hubbard model, in its original version $\hat{H} = -t \sum_{ij \sigma} \hat{c}^\dag_{i \sigma} \hat{c}_{j \sigma} + U \sum_i \hat{n}_{i \uparrow} \hat{n}_{i \downarrow}$, is a simplification of the reality thanks to a discretization of the space $r \in \mathrm{I\!R} \to i \in \mathrm{I\!I}$ and to a reduction of the range of the interaction, from the longest-range interaction in nature, the Coulombian $1/r$, to a short-range interaction $U \hat{n}_{i \uparrow} \hat{n}_{i \downarrow}$ local to a site $i$. It is true that the site $i$ can be thought both as an effective discretized point in space $r_i$, but also as a more spread-in-space localized orbital $\phi_i(r)$. However along this way the interaction and the model become dependent on the choice.of the basis set. Suppose that we have two electrons in two different localized orbitals $\phi_i(r)$ and $\phi_j(r)$: they won't interact if we choose as basis set of the Hubbard model the $\{\phi_i\}$, but they would interact if we choose another basis set, say Wannier functions $w_k(r)$: the two wavefunctions can both have a contribution on some same Wannier function, $\langle w_k | \phi_i \rangle \ne 0 \quad \langle w_k | \phi_j \rangle \ne 0$ and so interact. In principle the physics must not depend on the chosen basis-set, one should find the same result using different basis-set in the calculation. But here it is not the case, we have a model presenting a basis-set dependent physics.
Second and main point: the raison d'être of this model is the replacement of the many-body long-distance interaction by a local interaction (or an interaction local with respect to a given basis set $\phi_i$) in the hope to have a viable simplification of the many-body problem. Two electrons interact only if they are both in the same site $r_i$ or in the same state $\phi_i$, and do not interact at all if they are in two different sites or states $\phi_i$ and $\phi_j$. Even though the two sites are close or the distributions associated to the two orbitals $\phi_i , \phi_j$ are close in space or even overlap. Which is not the case in the real world. And it cannot become the case even by playing with the range/localization of the orbitals. One can extend the range of the interaction by making the model more complex and extending the interaction to next-neighbors sites/orbitals, but the long-range nature of the Coulomb interaction is such that a finite-rank of next-neighbours cannot really capture it. Note that, along the way to be more and more realistic, a model can become more complex than the reality.
On the other hand, it is true that the Hubbard model proved to be a good representation of many-body systems with short-range interaction, like for example cold atoms systems.
The Editor proposal for a Multiband Hubbard Monomer model to simulate the real helium atom is interesting indeed!
However, as above, how physical or unphysical this model turns out to be in particular question of: which interaction are you considering among the on-site multi bands? Are the electrons on different bands all interacting by the same strength U? Or are electrons interacting by U only if they are on the very same band with different spins? I also guess that the choice of the bands is not a marginal one. Probably, to simulate the helium atom, it would be more judicious to choose not really Bloch bands, and rather hydrogenic Z=2 single-particle atomic orbitals Anyway, I encourage the Editor to develop such model really in the spirit of amazement discussed above: this is interesting per se. However at the end a comparison on the excitation spectrum between the model and the exact Hylleraas solution, is unavoidable if she/he wishes to validate the model as a faithfull representation of the reality. Notice that the model can safely be validated only against an exact solution, that is a solution like Hylleraas that coincides to any level of significant digits to the experiment. Indeed, if we only had the experiment, the model could have found an excitation spectrum with a level very close to, say, the experimental level at 19.8 eV that Hylleraas unambiguosly interpreted as $2^3\!S$ state. But the model could have interpreted that level very differently, I don't know, for example as a lower Hubbard band!

---

## Round 3 · List of Changes

Summary of Changes:
Page 1: "analytic" changed (twice) to "closed-form" to correct an inaccuracy following Ref. 2 points 3 and 4.
Page 2: A paragraph introducing the Spherium model has been added together with the citation to a relevant reference, follwoing Ref. 2 point 5.
Page 2: The section introducing SCRPA and r-RPA has been dropped since too cryptic, following Ref. 2 point 6.
Page 2, Added footnote [26] at page 21, to discuss the story of gaussian basis-set adapted to DFT pure functionals or hybrids etc. and citations to the relevant references, following Ref. 1, point 5 and Ref. 2, point 2.
Page 3: Added all the Ref. 2 requested references for DFT-LDA+TDLDA, point 7.
Page 5: Added mathematical details about the characteristics of the Hylleraas-like series, following Ref. 1 point 1 and Ref. 2 point 3 and 4.
Page 5: Added details about modifications to calculate L>0 angular momentum He solutions, following Ref. 2 point 8.
Page 5, Sec. III: Wiggly sentence dropped, follwoing Ref. 2 point 10.
Page 5, Sec. III.A Specification that lambda runs over both singlet and triplet states, follwoing Ref. 2 point 11.
Page 7, Formulas for the ground-state correlation energies and text: Specification that the sums over lambda runs only over the positive (resonant) energies, following Ref. 1 point 2.
Page 8, Sec III.C: "computing power" -> "computing resources" (follwoing Ref. 2 point 15. And mentioning of other close to full CI methods, like CIPSI and CCSD, and relative citations, following Ref. 2 point 14.
Page 12:Sec IV.B: "exact DFT kinetic energy" -> "exact KS kinetic energy" following Ref. 1, point 6.
Page 13, Sec. IV.C: Paragraph introduced to discuss the mitigation of gaussian basis-set problems occurring in large molecules, following Ref. 1, point 5.
Page 15 and 16: Paragraph introduced to discuss the possibility of a scissor operator correction to the DFT-LDA+TDLDA spectrum, follwoing Ref. 1, point 7.

---

## Editorial Decision

published